# Subcortico-amygdala pathway processes innate and learned threats

Valentina Khalil[1,2,3†], Islam Faress[1,2,3,4*†], Noëmie Mermet-Joret[1,2,3†], Peter Kerwin[2], Keisuke Yonehara[1,4,5,6], Sadegh Nabavi[1,2,3*]

[1]Department of Molecular Biology and Genetics, Aarhus University, Aarhus, Denmark; [2]DANDRITE, The Danish Research Institute of Translational Neuroscience, Aarhus University, Aarhus, Denmark; [3]Center for Proteins in Memory – PROMEMO, Danish National Research Foundation, Aarhus University, Aarhus, Denmark; [4]Department of Biomedicine, Aarhus University, Aarhus, Denmark; [5]Multiscale Sensory Structure Laboratory, National Institute of Genetics, Mishima, Japan; [6]Department of Genetics, The Graduate University for Advanced Studies (SOKENDAI), Mishima, Japan

*For correspondence:
islam.faress@biomed.au.dk (IF);
snabavi@dandrite.au.dk (SN)

[†]These authors contributed equally to this work

Competing interest: The authors declare that no competing interests exist.

**Abstract** Behavioral flexibility and timely reactions to salient stimuli are essential for survival. The subcortical thalamic-basolateral amygdala (BLA) pathway serves as a shortcut for salient stimuli ensuring rapid processing. Here, we show that BLA neuronal and thalamic axonal activity in mice mirror the defensive behavior evoked by an innate visual threat as well as an auditory learned threat. Importantly, perturbing this pathway compromises defensive responses to both forms of threats, in that animals fail to switch from exploratory to defensive behavior. Despite the shared pathway between the two forms of threat processing, we observed noticeable differences. Blocking β-adrenergic receptors impairs the defensive response to the innate but not the learned threats. This reduced defensive response, surprisingly, is reflected in the suppression of the activity exclusively in the BLA as the thalamic input response remains intact. Our side-by-side examination highlights the similarities and differences between innate and learned threat-processing, thus providing new fundamental insights.

## Editor's evaluation

This study presents valuable insights into the circuits that are common for innate and acquired threats. The evidence supporting the conclusions is convincing, and the use of state-of-the-art methodology for the study of neural circuits, including chemogenetics, optogenetics, and fiber photometry, is appropriate. This work will be of interest to neuroscientists studying defensive behaviors as well as those in the field of multisensory thalamic integration.

## Introduction

Survival is the direct product of maximizing gains and avoiding harms. This is achieved by integrating brain circuitry that ensures survival through environment exploration while maintaining threat detection and avoidance (*Blanchard et al., 2001*; *Evans et al., 2019*; *Gross and Canteras, 2012*; *Headley et al., 2019*; *Orsini and Maren, 2012*; *Silva et al., 2016*). Such capacity can be acquired through learning, a mechanism widely believed to rely on synaptic plasticity (*Mongeau et al., 2003*; *Johansen et al., 2011*; *Quirk et al., 1995*; *Rogan et al., 1997*; *Tierney, 1986*). However, specific sensory stimuli could be innately appetitive or aversive and evoke approach or defensive behaviors, respectively (*Pereira and Moita, 2016*).

Rapid and continuous integration of innate and learned mechanisms promotes survival. This integration is ideally carried out as a timely and appropriate reaction to sensory stimuli. While cortical processing is necessary for cognitively demanding tasks, it is not well suited for rapid threat detection and response. However, subcortical processing serves as a neural shortcut that can crudely and rapidly elicit defensive behaviors, bypassing the more deliberate and intricate cortical processing. This efficient processing is partially explained by the fact that subcortical areas are among the earliest brain areas that receive sensory information (*Carr, 2015*; *McFadyen et al., 2020*; *Pessoa, 2008*; *Pessoa and Adolphs, 2010*).

It is well established that the subcortical circuit from the multisensory thalamus, lateral thalamus (LT) to the basolateral amygdala (BLA) is necessary for the acquisition and the recall of auditory learned threat conditioning in rodents (*Barsy et al., 2020*; *Edeline and Weinberger, 1992*; *Iwata et al., 1986*; *Lee et al., 2021*; *LeDoux et al., 1984*; *Romanski and LeDoux, 1992a*; *Romanski and LeDoux, 1992b*; *Romanski et al., 1993*; *Taylor et al., 2021*). However, the role of the subcortical LT-BLA pathway is underemphasized and understudied in processing innate threats (*Kang et al., 2022*).

Therefore, we sought to examine this pathway for processing innate threats as well, with the view that a side-by-side comparison may lead to a new mechanistic insight into the similarities and differences between circuits processing innate and learned threats.

Here, we show that, as with the learned threat conditioning, the LT-BLA pathway is essential for processing the innately aversive looming stimulus. Inactivation of either the BLA or the BLA-projecting neurons in the LT was sufficient to impair defensive responses to innate and learned threats. Additionally, fiber photometry from the BLA neurons or the LT axons projecting to the BLA showed a rapid rise in their activity to both forms of stimuli. However, the activity was reduced as mice showed habituation to the aversive stimuli. Despite similarities in processing the innate and learned threat, we found that propranolol, a β-adrenergic receptor blocker, specifically impairs the innate defensive response, while the response to the learned threat remains intact.

## Results

### BLA activity is required for processing a visual innately aversive threat and aversive conditioning

Threat conditioning is an associative learning paradigm where an initially neutral tone (conditioned stimulus [CS]) is repeatedly paired with an aversive footshock (unconditioned stimulus [US]). The next day, mice, upon exposure to the CS, show freezing responses (conditioned response [CR]), indicating successful learning of the association (*Blair et al., 2001*; *Pape and Pare, 2010*).

As for the innately aversive threat, we used the looming stimulus, an overhead expanding black disk that is thought to mimic an approaching aerial predator (*Yilmaz and Meister, 2013*). Unlike the tone used in threat conditioning, the looming stimulus triggers defensive responses without prior learning. The defensive response may vary from freezing to escapes and tail rattling (*Yilmaz and Meister, 2013*; *Salay et al., 2018*). In this study, we tested mice in an arena devoid of shelter (*Shang et al., 2018*). This setup promotes freezing as the dominant defensive response, comparable to the freezing response observed in threat conditioning. Although the mice showed extended freezing beyond the looming stimulus period, we observed rapid habituation to the repeated presentation of the loom (*Figure 1—figure supplement 1A–C*).

It is well established that the BLA activity is required to process learned threats (*Maren et al., 1996*; *Maren, 1999*; *Anglada-Figueroa and Quirk, 2005*; *Johansen et al., 2014*). To test whether the BLA is required for visually evoked innate defensive response, we applied a loss-of-function approach by transiently inactivating the BLA. We infected the BLA pyramidal neurons with hM4Di (Gi-coupled human muscarinic M4 designer receptor exclusively activated by a designer drug [iDREADD]) tagged with m-Cherry fluorescent proteins. We validated that the efficacy of iDREADDs' agonist, clozapine-N-oxide (CNO), mediated inhibition by performing optical stimulation of LT axons and in vivo electrophysiology recording in the BLA (*Figure 1A*). Intraperitoneal injection of CNO reduced the light-evoked BLA activity significantly (*Figure 1B* and *Figure 1—figure supplement 2A*; *Armbruster et al., 2007*; *Stachniak et al., 2014*). Behaviorally, the iDREADD-mediated BLA silencing significantly reduced the defensive responses of the animals to the looming stimulus (*Figure 1C and D* and *Figure 1—figure supplement 2B–G*).

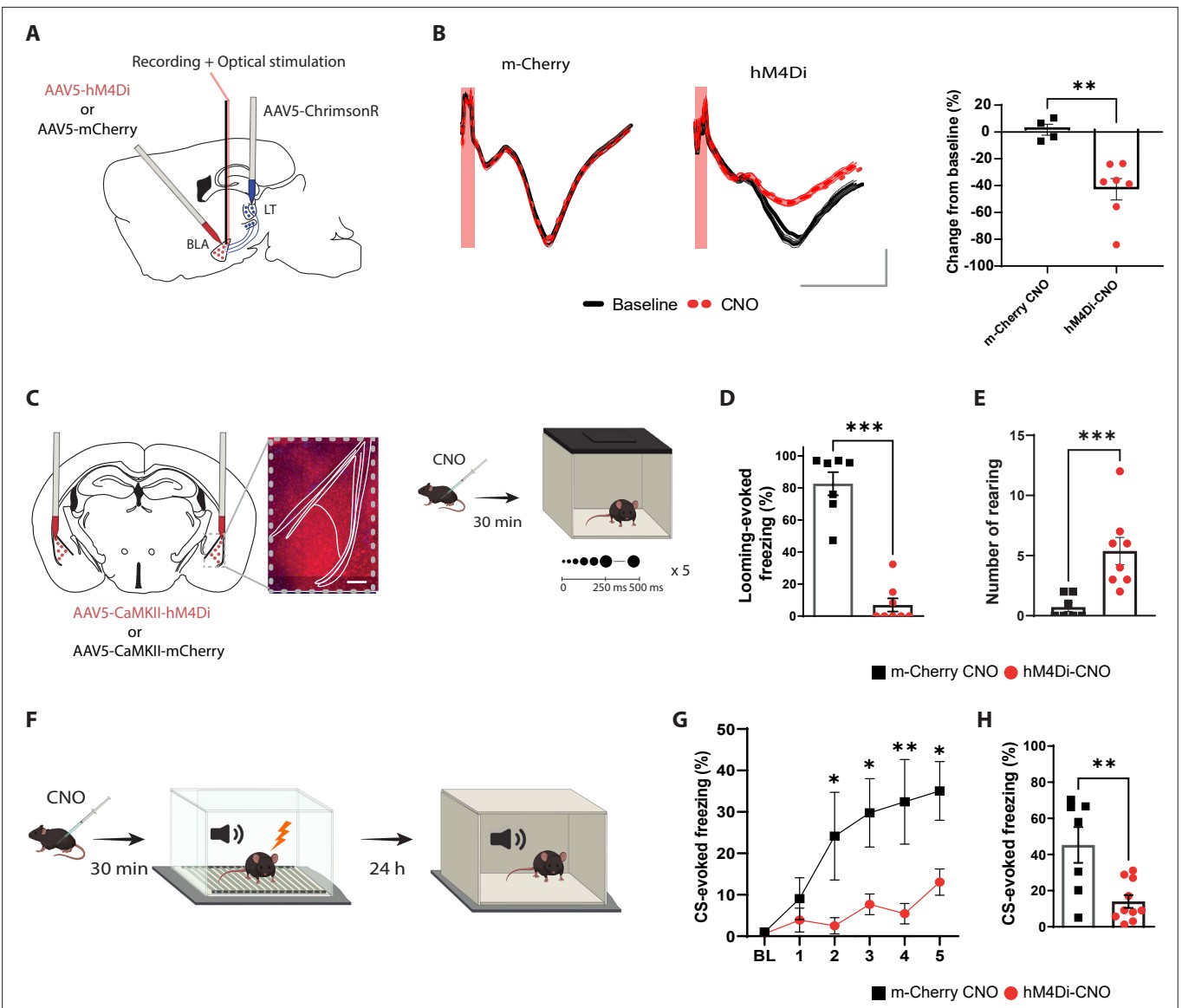

**Figure 1.** The basolateral amygdala (BLA) activity is required for processing innate as well as learned aversive signals. (**A**) Experimental design of the in vivo electrophysiology experiment. Mice were injected unilaterally with AAV vectors expressing ChrimsonR in the lateral thalamus (LT) and with hM4Di or m-Cherry in the BLA. (**B**) Clozapine-N-oxide (CNO) reduces the amplitude of the field excitatory postsynaptic potential (fEPSP) in hM4Di- but not m-Cherry-expressing neurons. Left panel: fEPSP is unchanged after CNO injection in the m-Cherry group. Middle panel: fEPSP is reduced after CNO injection in the hM4Di group. Shadowed area represents the SEM. Scale bar, 5 ms, 0.1 mV, the red bar represents the pulse of light (0.5 ms, 638 nm). Right panel: differential score comparing the change between before and after CNO injection in the two groups (m-Cherry-CNO, n = 4; hM4Di-CNO, n = 7; unpaired *t*-test, p-value=0.0033). (**C**) Experimental design of the behavioral experiment. Mice were injected bilaterally with AAV-expressing hM4Di or m-Cherry in the BLA. Scale bar, 250 um. After 3 wk of virus expression, the mice were exposed to the looming stimulus 30 min after CNO injection. (**D**) The freezing level is significantly reduced in the hM4Di-CNO group (n = 8) compared to the m-Cherry-CNO group (n = 7; Mann–Whitney test, p-value=0.0003). (**E**) The rearing events are significantly increased in the hM4Di-CNO group (n = 8) compared to the m-Cherry-CNO group (n = 7; Mann–Whitney test, p-value=0.0006). (**F**) One day after the looming exposure, the same animals were injected with CNO 30 min prior to the aversive conditioning protocol. Twenty-four hours later, the mice were tested in a new context in a CNO-free trial. (**G**) Freezing level during the baseline period (BL) and the five tone and foot-shock pairings. The conditioned stimulus (CS)-evoked freezing is significantly reduced in the hM4Di-CNO group (n = 10) than the m-Cherry-CNO group (n = 7; repeated-measures ANOVA for group by time interactions, *F*: 5,80 = 3.916, p-value=0.0032 with Sidak test correction). (**H**) The CS-evoked freezing in a new context is significantly reduced in the hM4Di-CNO group (n = 10) compared to the m-Cherry-CNO group (n = 7; unpaired *t*-test, p-value=0.0041). Results are reported as mean ± SEM. *p<0.05; **p<0.01; ***p<0.001.

The online version of this article includes the following figure supplement(s) for figure 1:

**Figure supplement 1.** The looming stimulus response is rapidly habituated.

*Figure 1 continued on next page*

*Figure 1 continued*

**Figure supplement 2.** The basolateral amygdala (BLA) is required for the looming stimulus defensive response.

**Figure supplement 3.** Lesioning basolateral amygdala (BLA) blocks the defensive responses to the looming stimulus and the aversive conditioning.

**Figure supplement 4.** Effect of the surgery on the looming stimulus-evoked responses.

Specifically, BLA inhibition resulted in failure of switching from rearing behavior to freezing, which is a typical looming stimulus-evoked behavior (*Figure 1E* and *Figure 1—figure supplement 2E-G*). The following day, the same mice were subjected to the threat conditioning protocol in the presence of CNO (*Figure 1F*). Consistent with previous reports (*Anglada-Figueroa and Quirk, 2005*; *Johansen et al., 2014*; *Maren, 1999*; *Maren et al., 1996*), transient inactivation of the BLA during the conditioning reduced the freezing response to the CS during the conditioning as well as the recall periods (*Figure 1G and H* and *Figure 1—figure supplement 2H and I*). Neither the injection of CNO nor the expression of iDREADDs alone interfered with the expression of the defensive responses to innately aversive and learned threats. In addition, mice with a permanent lesion in the BLA showed a similar deficit in defensive responses (*Figure 1—figure supplement 3*). Thus, the neuronal activity within the BLA is not only required for encoding and processing the learned threat but it is also essential for processing an innately aversive threat. This, to our knowledge, is the first direct evidence documenting that the BLA is required for processing an innate visual threat cue.

## The selective lesion of the BLA-projecting LT neurons impairs the defensive responses to the looming stimulus and the aversive conditioning

The subcortical pathway comprising the LT inputs to the BLA is known to be essential for the acquisition of threat conditioning (*Romanski and LeDoux, 1992a*; *Romanski and LeDoux, 1992b*; *LeDoux et al., 1984*; *Campeau and Davis, 1995*; *Barsy et al., 2020*; *Taylor et al., 2021*; *Lee et al., 2021*). Therefore, we tested whether these inputs are also required for processing innately aversive threats. We selectively lesioned BLA-projecting LT neurons by injecting retroAAV2-Cre in the BLA and a mixture of DIO-GFP and DIO-taCaspase3 in the LT (*Yang et al., 2013*; *Figure 2A*). Additionally, in two separate control groups, we injected retroAAV2-Cre in the BLA and DIO-GFP in the LT (GFP group) or a mixture of DIO-GFP and DIO-taCaspase in the LT (sham group). Our retrograde labeling was largely confined within the LT (*Figure 2B and D*).

Among mice with the selective lesion of the BLA-projecting LT neurons, we observed few GFP+ neurons (*Figure 2C, E and F*), demonstrating the efficiency of the approach. Furthermore, mice with the selective lesion showed a significant reduction in their defensive response to the looming stimulus as opposed to the two control groups (*Figure 2G* and *Figure 2—figure supplement 1A–C*). Similar to the BLA inhibition experiment, the selective lesion caused a similar failure in switching from exploratory to defensive behavior upon looming stimulus exposure (*Figure 2G* and *Figure 2—figure supplement 1A and B*). Likewise, the defensive responses to the learned aversive cue were reduced during the conditioning as well as the recall sessions (*Figure 2H* and *Figure 2—figure supplement 1D and E*). Together these data, in line with a recent report (*Kang et al., 2022*), demonstrate that the activity of the BLA-projecting LT neurons is required for the defensive responses not only to a learned but also to an innately aversive threat.

The BLA receives monosynaptic inputs from the temporal associative cortex (TeA) and auditory cortex (AuC). Since the LT is a major input to the TeA and the AuC, we tested whether the effect we observed in the selective lesion of the LT projecting cells could be an indirect effect via interrupting the cortical inputs to the BLA (*Barsy et al., 2020*). To this end, we lesioned the cortical regions (TeA and AuD) that are known to receive LT projections and in turn project to the BLA (*Figure 2—figure supplement 2A and B*). Mice with cortical lesions had a similar level of freezing in response to looming (*Figure 2—figure supplement 2C*) and conditioned stimuli (*Figure 2—figure supplement 2D and E*). Our results are in agreement with previous studies showing that selective optical inhibition of LT-BLA axons, which spares the disynaptic input from the LT-TeA/AuC-BLA, impairs defensive responses to the innate and learned threat stimuli (*Barsy et al., 2020*; *Kang et al., 2022*).

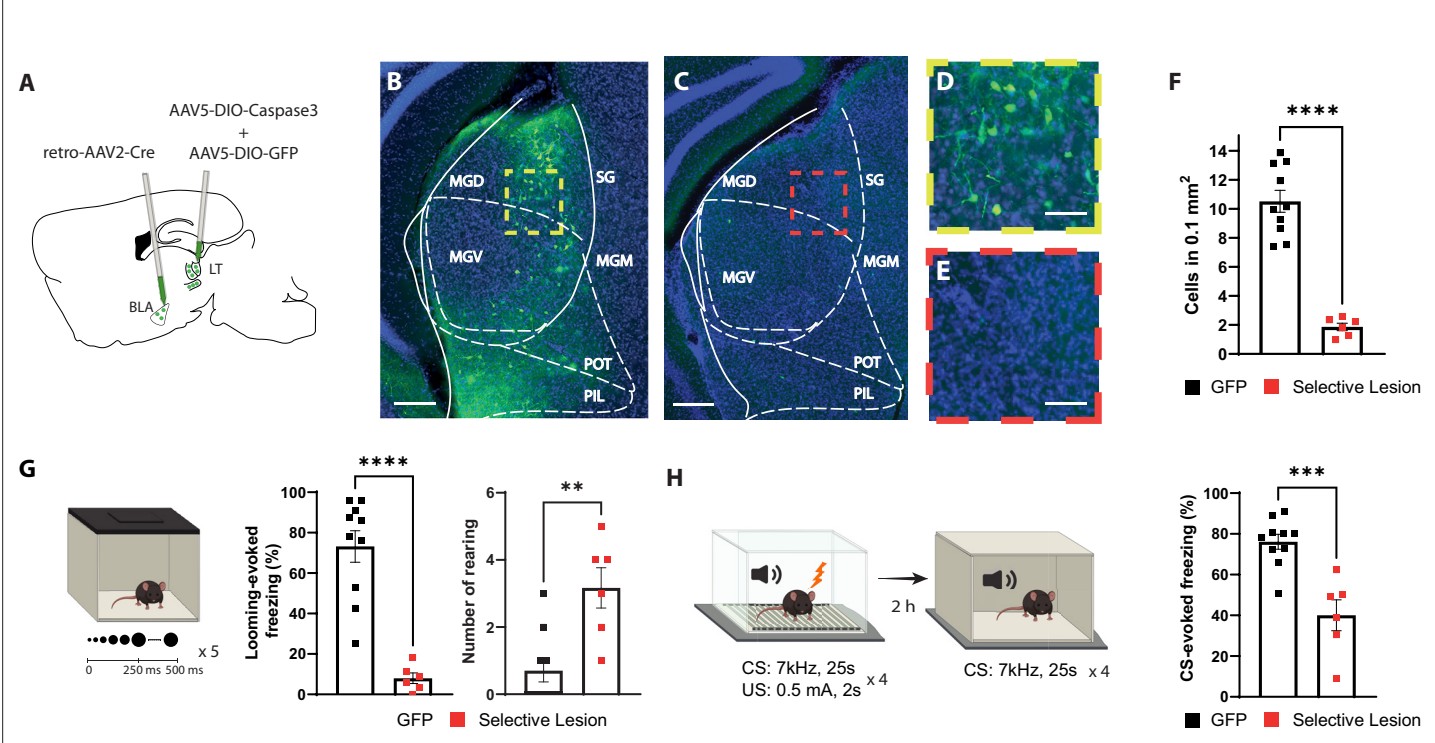

**Figure 2.** Selective lesion of the basolateral amygdala (BLA)-projecting neurons in the lateral thalamus (LT) impairs the defensive responses to the looming stimulus and the aversive conditioning. (**A**) Viral strategy for the selective lesion of BLA-projecting LT neurons. The GFP group was injected with retro-AAV2-Cre in BLA and AAV5-DIO-GFP only in the LT. (**B, C**) Representative image showing retrogradely transported GFP in the BLA-projecting LT neurons in a mouse from the GFP group (**B**) and a mouse from the selective lesion group (**C**), respectively. Scale bar, 200 um. (**D, E**) Zoomed-in images from the regions outlined in yellow from (**B**) and in red from (**C**). Scale bar, 50 um. (**F**) Quantification of GFP+ neurons in the LT. The selective lesion group (n = 6) showed a significant reduction in the number of GFP+ neurons compared to the GFP group (n = 10; unpaired *t*-test, p-value<0.0001). (**G**) Mice were exposed to the looming stimulus. The freezing level is significantly reduced in the selective lesion group (n = 6) compared to the GFP group (n = 10; Mann–Whitney test, p-value<0.0001). The rearing frequency is significantly higher in the selective lesion group (n = 6) compared to the GFP group (n = 10; Mann–Whitney test, p-value=0.0050). (**H**) The same mice were conditioned 1 d after the looming exposure. The mice were tested for memory recall in a new context 2 hr later. The selective lesion group (n = 6) showed a significant reduction in conditioned stimulus (CS)-evoked freezing compared to the GFP group during the short-term memory (STM) recall (n = 10; unpaired *t*-test, p-value=0.0003). Results are reported as mean ± SEM. **p<0.01; ***p<0.001; ****p<0.0001.

The online version of this article includes the following figure supplement(s) for figure 2:

**Figure supplement 1.** The selective lesion of the basolateral amygdala (BLA)-projecting neurons in lateral thalamus (LT) impairs the defensive responses to the looming stimulus and the aversive conditioning.

**Figure supplement 2.** Lesioning the temporal associative cortex (TeA) and auditory cortex (AuC) does not affect the defensive responses to the looming stimulus and aversive conditioning.

## The axons of the BLA-projecting LT neurons are activated by the looming stimulus and show an increase in CS-evoked response following aversive conditioning

In the preceding section, we demonstrated that the activity of the BLA-projecting LT neurons is essential for processing innate and learned threats. Therefore, we expect an increase in the activity of the LT input to the BLA that is time-locked to the threat signals. For this purpose, we took advantage of fiber photometry in freely moving mice. Virus expressing the genetically encoded Ca²⁺ indicator GCaMP7s (*Dana et al., 2019*) was injected into the LT, and a fiber optic was implanted above the dorsal tip of the BLA (*Figure 3A–C* and *Figure 3—figure supplement 1A*). The axonal activity of the LT neurons serves as a proxy for their release of neurotransmitters into the BLA. The time-locked GCaMP activity of the LT projections to the onset of the looming stimulus was evident. As mice showed habituation to the stimuli, GCaMP activity diminished, with later stimuli eliciting neither defensive behavior nor time-locked GCaMP activity in the LT inputs (*Figure 3D–J*).

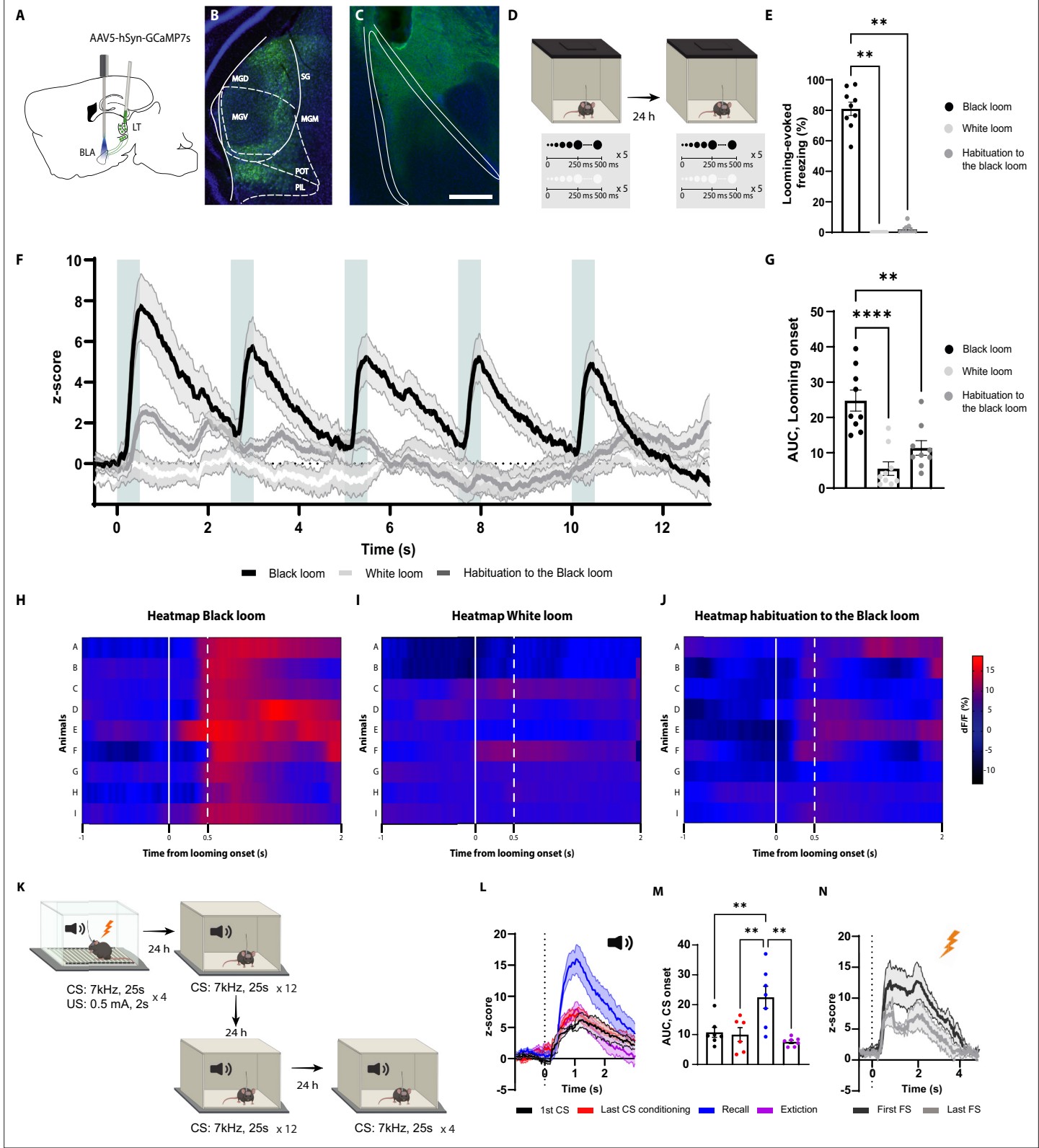

**Figure 3.** The axons of the basolateral amygdala (BLA)-projecting neurons in the lateral thalamus (LT) are activated by the looming stimulus and show an increase in conditioned stimulus (CS)-evoked response following aversive conditioning. (**A**) Illustration showing virus injection and the optic fiber implantation strategy for GCaMP7s recordings from the axon terminals of the BLA-projecting MGN neurons. (**B**) Representative image of GCaMP7s expression in the MGN. Scale bar, 200 um. (**C**) Representative image of the optic fiber location above the BLA. Scale bar, 500 um. (**D**) After 5 wk of virus

*Figure 3 continued on next page*

*Figure 3 continued*

expression, the mice were exposed to the black and white looming stimulus. Twenty-four hours later, the mice were reexposed again to the looming stimulus. (**E**) Freezing level to the black looming stimulus (n = 9), to the white looming stimulus (n = 9), and after the habituation to the black looming (n = 9, Friedman test p-value<0.0001 with Dunn's test). (**F**) Z-score of the $Ca^{2+}$ response from the MGN axon terminals during the black looming stimulus (n = 9; in black; time-to-peak, mean: 818 ms, SEM: ±114 ms), white looming stimulus (n = 9; in white), and the habituation to the black looming stimulus (n = 9; in gray). (**G**) The area under the curve (AUC) is significantly reduced when the mice are habituated to the black looming compared to the first exposure to it (n = 9; paired *t*-test, p-value=0.0001). (**H–J**) Heatmap of the response to the first expansion of the black looming stimulus (**H**), white looming stimulus (**I**), and after the habituation to the black looming stimulus (**J**) for each mouse. (**K**) Mice were conditioned and tested as previously described. After the recall session, the same went through an extinction protocol for the following two days. (**L**) Z-score of the $Ca^{2+}$ response from the axon terminals of the BLA-projecting MGN neurons during the first CS presentation (n = 7; in black; time-to-peak, mean: 973 ms, SEM: ±160 ms), the last CS presentation of the conditioning (n = 6; in red; time-to-peak, mean: 939 ms, SEM: ±62 ms), the first CS presentation during the recall session (n = 7; in blue; time-to-peak, mean: 948 ms, SEM:±41 ms), and the first CS presentation after the extinction training (n = 7; in purple; time-to-peak, mean: 956 ms, SEM: ±96 ms). (**M**) The AUC is significantly increased when the mice are exposed to the CS during the recall session (n = 7) compared to the CS-evoked response at the beginning of the conditioning (n = 7) and at the end of the conditioning (n = 7) and after the extinction training (n = 7; mixed-effects analysis, *F*: 3,17 = 8.791, p-value=0.0010 with Tukey test correction). (**N**) Z-score of the $Ca^{2+}$ response from the axon terminals of the BLA-projecting neurons in the MGN during the first (time-to-peak, mean: 776 ms, SEM: ±57 ms) and last footshock presentation (n = 5; time-to-peak, mean: 660 ms, SEM: ±76 ms). Results are reported as mean ± SEM. **p<0.01; ***p<0.001.

The online version of this article includes the following figure supplement(s) for figure 3:

**Figure supplement 1.** Optic fiber location and efficacy of extinction protocol.

Notably, the white looming stimulus, as opposed to a black looming stimulus, did not evoke defensive behavior (*Yilmaz and Meister, 2013*), nor did it trigger GCaMP activity (*Figure 3E–I*). This, along with the result from the habituation sessions, indicates that the increased activity of the LT inputs is not merely the product of the sensory property of the looming stimulus, but it reflects the saliency and aversiveness of the stimulus, as well.

We next monitored the activity of the LT inputs during aversive conditioning, the recall sessions, and post-extinction training (*Figure 3K*). As expected from a multisensory brain region (*Bordi and LeDoux, 1994*; *Linke et al., 1999*; *Linke, 1999*), the LT inputs were activated from the tone onset (*Figure 3L and M*). The amplitude of the activity remained unchanged for the subsequent CS presentations, despite mice showing an increased CS-evoked freezing to these stimuli (*Figure 3L*). Interestingly, during the recall session 24 hr later, we observed a significant increase in CS-evoked GCaMP activity, which was not evident during the conditioning (*Figure 3L and M*). In addition, upon extinction training, the CS-evoked activity returned to its preconditioning level (*Figure 3L and M* and *Figure 3—figure supplement 1B*). Moreover, the footshock induced a time-locked increase in GCaMP activity, which was reduced in amplitude with subsequent US delivery (*Figure 3N*). From these experiments, we conclude that the LT neurons directly convey the signals for innately aversive as well as for learned threats to the BLA.

## Contralateral disconnection of the LT-BLA pathway impairs the defensive responses to the looming stimulus and the aversive conditioning

Our inactivation experiments demonstrate that the BLA (*Figure 1*) as well as the BLA-projecting neurons in the LT (*Figure 2*) are necessary for the processing of the learned and innate aversive threat responses. However, the previous experiments on their own cannot distinguish whether the two regions function in series, with the BLA receiving the threat signals from the LT (as indicated by GCaMP activity in the LT inputs) (*Figure 3*); or, the LT and the BLA function in parallel, with the BLA receiving the signal from other sources. To address this issue, we used an asymmetrical disconnection approach (*LeDoux et al., 1986*; *Iwata et al., 1986*; *Eldridge et al., 2016*; *Barker et al., 2017*; *Torromino et al., 2019*), where we inhibited the activity of the LT and the BLA contralaterally. This approach is suited to test whether the direct connection between two regions is required for a particular function (*Figure 4A and B*). The prerequisite is the connections should be ipsilateral and not reciprocal, as it is the case for the LT and the BLA (*LeDoux et al., 1986*; *LeDoux et al., 1990*).

For this purpose, we applied a reversible disconnection between the LT-BLA pathway by expressing hM4Di in the LT and the BLA contralateral to each other (*Figure 4A and B* and *Figure 4—figure supplement 1A and B*). This manipulation spared the lateral geniculate nucleus (LGN) (*Figure 4—figure*

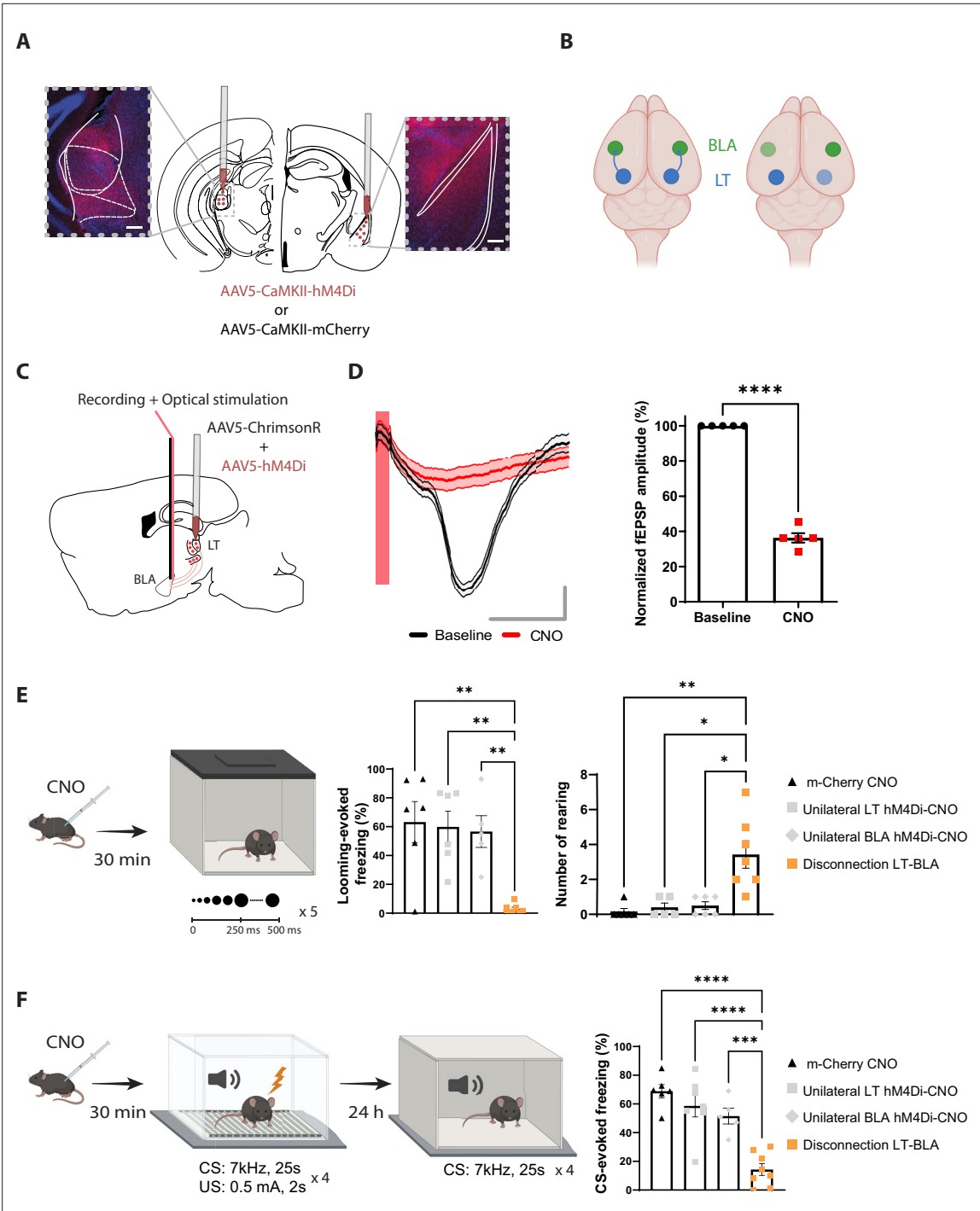

**Figure 4.** Reversible contralateral disconnection of the lateral thalamus-basolateral amygdala (LT-BLA) pathway impairs the defensive responses to the looming stimulus and the aversive conditioning. (**A**) hM4Di injections in the LT and the contralateral BLA. Scale bar, 200 um. (**B**) Left: the LT and the BLA are connected through a non-reciprocal and ipsilateral connection. Right: contralateral disconnection of the LT-BLA pathway. (**C**) Diagram showing the experimental design of the in vivo electrophysiology experiment. Mice were co-injected unilaterally with AAV vectors expressing ChrimsonR and hM4Di in the LT. (**D**) Clozapine-N-oxide (CNO) reduced the field excitatory postsynaptic potential (fEPSP) in mice expressing hM4Di and injected with CNO. Left panel: representative traces from one mouse from the hM4Di-CNO group. The red bar represents the pulse of light (0.5 ms, 638 nm). Shadowed area represents SEM. Scale bar, 5 ms, 0.1 mV. Right panel: the graph shows the normalized fEPSP values before and after CNO injection in mice expressing hM4Di and injected with CNO (n = 5; paired *t*-test, p-value<0.0001). (**E**) Mice were injected with CNO 30 min before being exposed to the looming stimulus. The disconnection LT-BLA group (n = 7) showed a significant reduction in the freezing level compared to all the groups. The unilateral inhibition of the LT (n = 6) and the BLA (n = 5) did not impair the looming stimulus-evoked freezing (ordinary one-way ANOVA, *F* = 3,20, p-

*Figure 4 continued on next page*

*Figure 4 continued*

value=0.0006). Right: the rearing events are significantly higher in the disconnection LT-BLA group (n = 7) compared to all the other groups during the looming stimulus presentation (Kruskal–Wallis test, *F* = 2,24, p-value=0.0016). (**F**) Mice were reinjected with CNO 30 min before aversive conditioning, and they were exposed to the conditioned stimulus (CS) in a new context in a CNO-free trial. The disconnection LT-BLA group (n = 7) showed a significant reduction in the CS-evoked freezing level compared to all control groups during the LTM recall (ordinary one-way ANOVA, *F* = 3,22, p-value<0.0001). Results are reported as mean ± SEM. *p<0.05; **p<0.01; ***p<0.001; ****p<0.0001.

The online version of this article includes the following figure supplement(s) for figure 4:

**Figure supplement 1.** Virus expression and maximal spreading in mice from *Figure 4*.

**Figure supplement 2.** Reversible and irreversible contralateral disconnection of the lateral thalamus-basolateral amygdala (LT-BLA) pathway impairs the defensive responses to the looming stimulus and the aversive conditioning.

*supplement 1C*). Electrophysiologically, upon CNO injection, the responses of the BLA neurons to the optical stimulation of LT neurons co-expressing ChrimsonR and hM4Di was greatly reduced (*Figure 4C and D*). Behaviorally, CNO-induced inactivation of the contralateral regions significantly reduced the defensive responses to the looming stimulus (*Figure 4E* and *Figure 4—figure supplement 2B-H*) and the recall session of the aversive conditioning (*Figure 4F* and *Figure 4—figure supplement 2I and J*). The locomotor activity and the anxiety-like behaviors, as measured by line crosses and time spent in the center of the looming arena, however, remained intact (*Figure 4—figure supplement 2A*). CNO injection in mice expressing mCherry in the contralateral LT and BLA did not reduce freezing responses in either of the behavioral tasks (*Figure 4E and F*). More importantly, unilateral inactivation of the LT and the BLA was not sufficient to block the defensive responses (*Figure 4E and F*). Similar results were obtained when we performed an irreversible disconnection of the LT-BLA pathway (*Figure 4—figure supplement 2K–N*). These experiments suggest that the direct projection from the LT to the BLA is required for the processing of innately aversive threats as well as learned threats.

## The LT lesion blocks the BLA neuronal activation by the looming and conditioned stimuli as well as reduces the response to a footshock

The disconnection of the LT-BLA pathway impairs the defensive responses to both forms of threat cues (*Figure 4*), suggesting that this pathway is the main root by which the BLA receives the aversive signals. If so, upon the LT lesion, the BLA responses to the aversive stimuli must largely disappear. We co-injected AAV vectors expressing DIO-ta-Capsase3 and Cre recombinase in the LT and GCaMP8m in the BLA (*Zhang et al., 2021*). GCaMP signal was collected through a fiber optic implanted above the tip of the BLA (*Figure 5A* and *Figure 5—figure supplement 1A*). The control group underwent the same procedure except that no Cre recombinase was injected.

In the LT-lesioned mice, the BLA response to the black looming stimuli, along with the behavioral defensive responses, largely disappeared (*Figure 5B–F*). In the non-lesioned mice, where defensive responses to the looming stimuli remained intact, we observed a timed-locked GCaMP activity to the stimuli in the BLA (*Figure 5B–E*). After habituation, the behavioral and neuronal responses to the black looming stimulus in the control group were comparable to those lesioned (*Figure 5B–G*). Moreover, the white looming stimulus did not produce a noticeable activation of the BLA neurons, nor did it elicit a defensive response (*Figure 5—figure supplement 1B–D*).

We next monitored the activity of the BLA during the auditory threat conditioning and the recall in the LT-lesioned and non-lesioned mice (*Figure 5H*). The LT-lesioned mice, as expected, did not produce a conditioning response to the tone (*Figure 5—figure supplement 1F and G*). Accordingly, the CS failed to activate the BLA in these mice (*Figure 5I–K* and *Figure 5—figure supplement 1E*), while the US-evoked response was significantly reduced (*Figure 5L*). In the non-lesioned mice, on the other hand, we observed tone-evoked response in the BLA, but with a latency that cannot be fully explained by the slow kinetics of the calcium indicator. As the conditioning progressed, we observed the appearance of an additional, smaller and shorter latency tone-evoked component. In the recall session, the tone-evoked response in the BLA was significantly larger in amplitude and shorter in latency than observed at the end of the conditioning session in the previous day (*Figure 5J*). Upon extinction, the tone-evoked response, along with the defensive behavior, was significantly reduced (*Figure 5I and J* and *Figure 5—figure supplement 1G*).

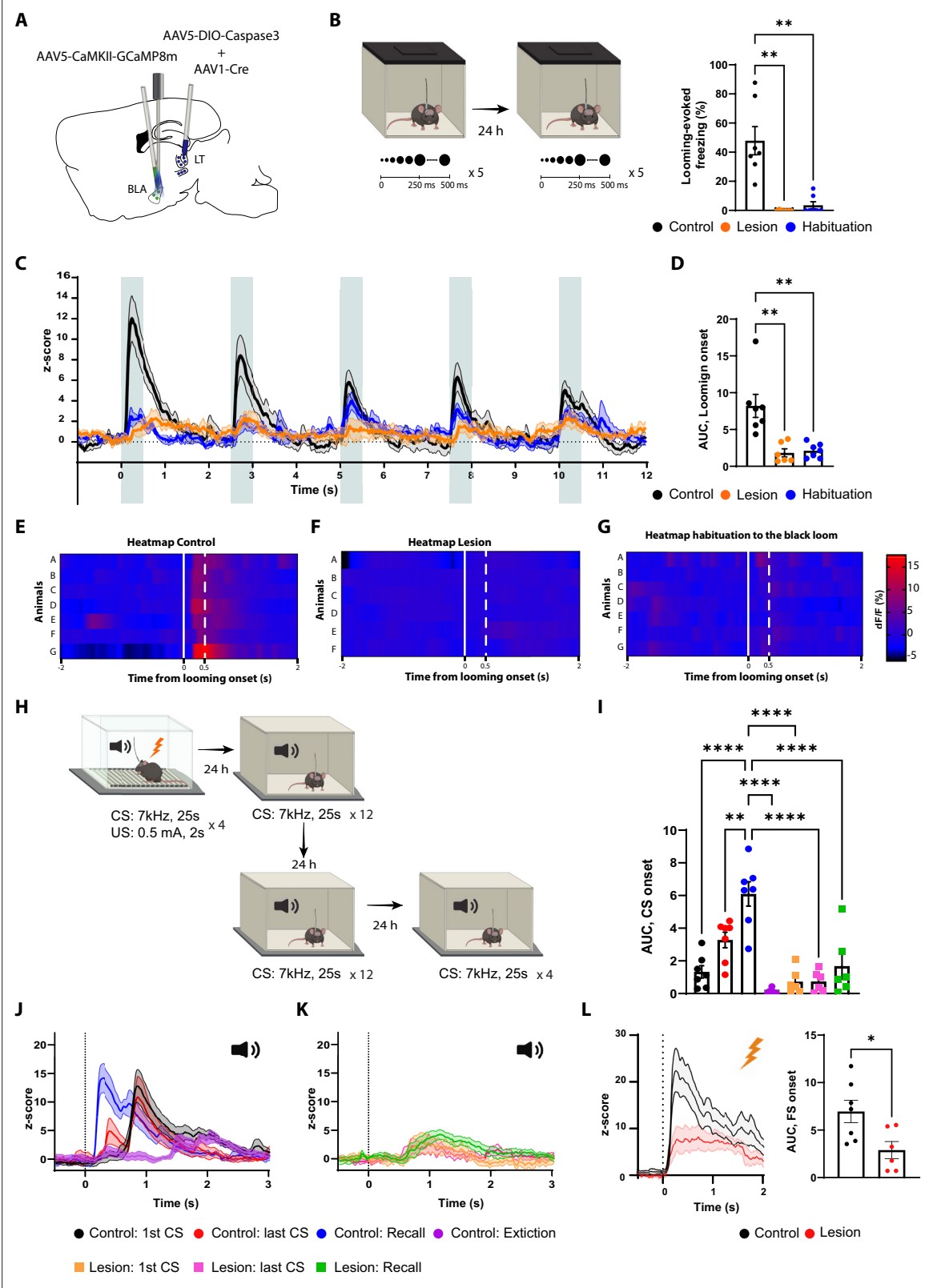

**Figure 5.** The lateral thalamus (LT) lesion impairs the basolateral amygdala (BLA) response to the looming, the conditioned stimulus (CS), and the unconditioned stimulus (US) stimuli. (**A**) Illustration showing the virus injections and the optic fiber implantation. (**B**) Mice were exposed to the looming stimulus on day 1. On day 2, the control group was habituated to the looming stimulus. The mice from the lesion group (n = 6) have a significant reduction in the freezing level to the looming stimulus compared to the control group (n = 7). Similarly to the control group, once the mice were

*Figure 5 continued on next page*

*Figure 5 continued*

habituated to the black looming stimulus (n = 7; Kruskal–Wallis test, F = 3,20, p-value<0.0001). (**C**) Z-score of the calcium response during the black looming stimulus in the control group (n = 7; in black; time-to-peak, mean: 245 ms, SEM: ±14 ms) and in the lesion group (n = 6; in orange) and the control group after the habituation to the black loom (n = 7; in blue). (**D**) The area under the curve (AUC) is significantly reduced in the lesion (n = 6) and in the habituation to the black loom (n = 7) groups compared to the control group (n = 7; ordinary one-way ANOVA, $F_{(2,17)}$ = 12.73, p-value=0.0004) during the exposure to the black looming stimulus. (**E–G**) Heatmap representing the individual response to the first expansion to the black looming stimulus for the control group (**E**) and for the lesion group (**F**), and for the control group after habituation to the black loom (**G**) for each mouse. (**H**) Mice were conditioned and tested as previously described. After the recall session, the same went through an extinction protocol for the following 2 d. (**I**) The AUC for the first CS presentation during the recall in the control group is significantly increased compared to all conditions of the lesion group (two-way ANOVA, F: 6,33 = 16.29, p-value<0.0001 with Tukey test correction). (**J**) Z-score of the calcium responses during the first (n = 7; time-to-peak, mean: 978 ms, SEM: ±153m s) and last CS (n = 7; the first time-to-peak, mean: 750 ms, SEM: ±72 ms) presentation during the conditioning (in black and in red), and during the first CS presentation during the recall (n = 7; in blue; time-to-peak, mean: 448 ms, SEM: ±85 ms) and after extinction training (n = 7; in purple; time-to-peak, mean: 1600 ms, SEM: ±240 ms) in the control group. (**K**) Z-score of the calcium responses during the first (n = 6) and last CS (n = 6) presentation during the conditioning (in orange and magenta), and during the first CS presentation during the recall (n = 6; in green) in the lesion group. (**L**) Left: Z-score of the $Ca^{2+}$ response during the footshock presentation for the control group (n = 7, in black; time-to-peak, mean: 387 ms, SEM: ±93 ms) and the lesion group (n = 6, in red). Right: the AUC is significantly reduced in the mice from the lesion group (n = 6) compared to the mice from the control group (n = 7; unpaired t-test, p-value=0.0219). Results are reported as mean ± SEM. *p<0.05; **p<0.01; ****p<0.0001.

The online version of this article includes the following figure supplement(s) for figure 5:

**Figure supplement 1.** The basolateral amygdala (BLA) is not activated by the white looming stimulus and the lateral thalamus (LT) lesion impairs the conditioned stimulus (CS)-evoked BLA response.

## Blocking β-adrenergic receptors reduces the defensive response to the innately aversive threat

Previous studies have shown that innately aversive stimuli such as fox urine (*Hu et al., 2007*; *Liu et al., 2010*) or cat fur odor (*Do Monte et al., 2008*) mediates defensive responses through the activation of the β-adrenergic receptor. Therefore, we considered that the threat response triggered by looming stimulus may rely on the activity of these receptors. To test this, prior to exposure to the looming stimulus mice were injected with propranolol, a β-adrenergic receptor blocker (*Figure 6—figure supplement 1A–C*). The defensive responses to the looming stimulus in these mice largely disappeared, while the exploratory behavior remained intact (*Figure 6—figure supplement 1B*). On the other hand, sotalol, a peripherally acting β-adrenergic blocker, had no impact on defensive responses, suggesting that propranolol reduced the looming stimulus-evoked defensive responses by acting on the central nervous system (*Figure 6—figure supplement 1B*).

To further examine the contribution of the subcortical pathway in the processing of innate threats, we injected propranolol in mice expressing GCaMP in either LT axons projecting to the BLA or in the BLA pyramidal neurons, followed by repeated exposure to looming stimuli (*Figure 6A* and *Figure 6—figure supplement 1K*). At the cellular level, looming stimuli failed to elicit a significant increase in the BLA activity, which reflects the reduced freezing response in mice injected with propranolol (*Figure 6B–D*). Surprisingly, the stimulus-evoked LT axonal activity remained undisturbed despite the lack of defensive response (*Figure 6E–G*). Of note, 48 hr later, when these mice were retested, defensive response, as well as time-locked BLA activity to the looming stimulus, was restored (*Figure 6B–E*).

To our surprise, propranolol did not impair the aversive conditioning responses when injected either before the conditioning or before the recall session (*Figure 6—figure supplement 1D–I*). In addition, propranolol injections prior to the recall session perturbed neither BLA (*Figure 6—figure supplement 2A–D*) nor LT axonal activity (*Figure 6—figure supplement 2E–G*).

## Discussion

The thalamic-BLA pathway and its intricate microcircuitry have been described for their function in processing the auditory CS and aversive US (*Barsy et al., 2020*; *Janak and Tye, 2015*; *Rogan et al., 1997*; *Rogan et al., 2005*). Yet, their role in processing unimodal innate threats remains under investigated (*Kang et al., 2022*). Here, we demonstrate that transient or permanent inactivation of the BLA (*Figure 1*) or BLA-projecting LT neurons (*Figure 2*) not only impairs threat learning but also abolishes all the measured defensive responses to an innate threat. More specifically, upon exposure to an innate visual threat, animals with a compromised LT-BLA pathway fail to switch from exploratory to defensive

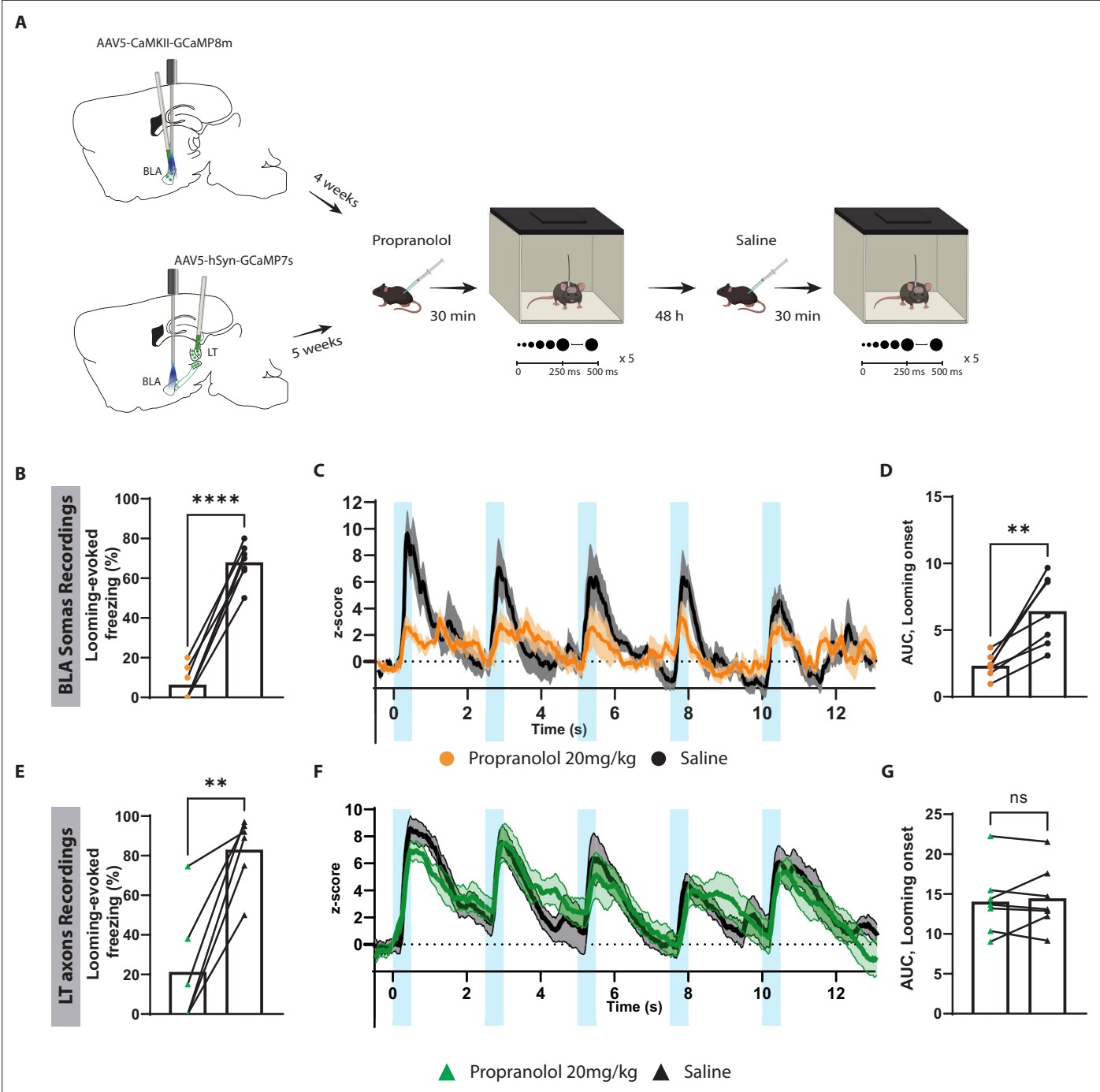

**Figure 6.** Blocking β-adrenergic receptors reduces the defensive response and the basolateral amygdala (BLA) activation to the innately aversive threat. (**A**) Timeline showing the different stages of the experiment. (**B**) The looming stimulus-evoked freezing is significantly reduced in the propranolol trial compared to the re-exposure trial in which the same mice were injected with saline (n = 7; paired *t*-test, p-value<0.0001). (**C**) The graph shows the average of the Z-score of the Ca²⁺ responses of the looming stimulus presentation in the BLA after propranolol injection (n = 7, in orange) and saline injection (n = 7; in black). (**D**) The area under the curve (AUC) is significantly reduced in the propranolol trial (n = 7) compared to the saline trial (n = 7; paired *t*-test, p-value=0.0034). (**E**) The looming stimulus-evoked freezing is significantly reduced in the propranolol trial compared to the re-exposure trial in which the same mice were injected with saline (n = 7; paired *t*-test, p-value=0.0022). (**F**) The graph shows the average of the Z-score of the Ca²⁺ responses of the looming stimulus presentation of the lateral thalamus (LT) axon terminals after propranolol injection (n = 7, in green) and saline injection (n = 7; in black). (**G**) The AUC is unchanged in the propranolol trial (n = 7) compared to the saline trial (n = 7; paired *t*-test, p-value=0.6016). Results are reported as mean ± SEM. ns, nonsignificant; **p<0.01; ****p<0.0001.

The online version of this article includes the following figure supplement(s) for figure 6:

*Figure 6 continued on next page*

*Figure 6 continued*

**Figure supplement 1.** Propranolol reduces defensive response to a looming stimulus but it does not impair aversive conditioning.

**Figure supplement 2.** Propranolol does not affect the conditioned stimulus (CS)-evoked activity in the basolateral amygdala (BLA) or in the lateral thalamus (LT) projections to the BLA.

behaviors, displaying neither freezing nor escape reaction (*Figures 1, 2 and 4*). Furthermore, the LT axons projecting to the BLA (*Figure 3F*) and the neurons within the BLA produce time-locked activity to each looming stimulus in mice showing defensive responses (*Figure 5C*). With repeated exposure to looming stimuli, the neuronal activity, along with defensive responses, gradually fades. The habituation to looming stimuli was rapid (within-session) and long-lasting (for the entire duration of the experiment) (*Figures 3 and 5*). A recent study has shown that upon repeated exposure to a looming stimulus neuronal response within deeper layers of superior colliculus (SC) undergo rapid visual habituation. A form of short-term depression has been proposed to underlie the fast habituation (*Lee et al., 2020*). This phenomenon may underlie the within-session habituation that we observed here. This is consistent with the gradual reduction in the activity of the BLA-projecting neurons within LT, a region downstream of the deep layers of the SC (*Benavidez et al., 2021*; *Linke et al., 1999*). The long-lasting habituation, on the other hand, cannot be explained by such a visual adaptation or habituation that lasts on the order of seconds to minutes and not days (*Boehnke et al., 2011*; *Lee et al., 2020*; *Wark et al., 2009*). The long-term form of the habituation largely occurs upstream of the BLA as observed in the reduced activity in the LT projections to the BLA (*Figure 3F and G*). If, as we suspect, the signal for the looming stimulus to the LT originates from the SC (see below), the long-term form of the habituation may, at least in part, be processed within the LT. Further studies are necessary to test this possibility.

Although reduced defensive response to an innately aversive signal such as a looming stimulus has been categorized as habituation, we observed physiological similarities between habituation to a looming stimulus and extinction to a cued conditioning stimulus. Both phenomena reduce the heightened evoked activity of the LT projections as well as the BLA neurons. Additionally, while mice injected with propranolol did not show defensive response to a looming stimulus, 2 d later in the absence of the drug the same mice showed robust freezing response to the stimulus. This is similar to a study reporting that mice injected with propranolol before a cued fear recall have reduced freezing response to the cue, while in the following day the mice show a heightened freezing response (*Leal Santos et al., 2021*); unlike this study, however, we did not observe a physiological or behavioral effect of propranolol on cued conditioning (see below for discussion). The cellular and circuit mechanisms of extinction have been extensively studied (*Herry et al., 2006*; *Orsini and Maren, 2012*). These studies may provide mechanistic insight into the habituation that we have observed here. Further investigations in this line may have fundamental and translational value.

An overlap in processing innate and learned threats is not unique to the LT-BLA pathway. The lateral habenula and the central amygdala control the processing of a number of innate and learned threats (*Fadok et al., 2017*; *Lecca et al., 2017*; *Lecca et al., 2020*; *Matsumoto and Hikosaka, 2007*; *Root et al., 2014*; *Mondoloni et al., 2022*; *Tovote et al., 2016*; *Sachella et al., 2022*; *Isosaka et al., 2015*). Since both types of threats share a similar repertoire of defensive responses, such as freezing and escaping, the wiring economy favors the layout where there is closer physical proximity between the two circuits, as we observed here (*Klyachko and Stevens, 2003*; *Stevens, 2012*).

Direct and indirect evidence shows that the LT-BLA pathway processes other forms of innate threats as well. Recently, studies have demonstrated that inactivation of thalamic-BLA pathway reduces freezing response to intense sound and looming stimuli (*Kang et al., 2022*), as well as innately aversive ultrasound activates the BLA (*Mongeau et al., 2003*; *Shukla and Chattarji, 2022*). Moreover, the use of live predators as an innate threat has further supported this notion by showing that the BLA lesion in rodents eliminates defensive responses to the threat (*Bindi et al., 2018*; *Martinez et al., 2011*).

Since the LT is widely regarded as an auditory relay region (*Rogan and LeDoux, 1995*; *Rogan et al., 1997*; *Weinberger, 2011*), its role as the main source of innately aversive visual signal to the BLA may seem unexpected. However, it has been known that the LT receives auditory, somatosensory, visual, and multimodal information (*Bordi and LeDoux, 1994*; *Linke et al., 1999*; *Linke, 1999*). The lateral posterior nucleus of the thalamus (LP) has been proposed as another direct source conveying

a looming stimulus signal to the BLA (*Wei et al., 2015*). Our reversible disconnection of the LT-BLA pathway, which largely spares the LP, argues otherwise. The LT lesion not only abolishes the defensive responses to the looming stimulus but also largely eliminates the stimulus-evoked activity in the BLA (*Figure 5*).

Although the input source of the looming stimulus to the LT is unknown, we speculate the SC could be a likely candidate as it encodes threat and escape behavior to looming stimuli (*Evans et al., 2018*) and sends monosynaptic connections to the LT (*Linke et al., 1999*). Additionally, it has been shown that the SC may trigger defensive responses to the looming stimulus by conveying the signals to the periaqueductal gray (PAG) (*Evans et al., 2018*) or indirectly to the central nucleus of the amygdala (CeA) (*Shang et al., 2015*; *Zhou et al., 2019*). It is relevant to note that the BLA conveys learned aversive signals directly to the CeA (*Fadok et al., 2017*) and indirectly through the CeA to the PAG (*Tovote et al., 2016*). It will be of particular interest to test whether the information signaling the innate threat is communicated through the same channel. As the SC and the BLA, the two essential regions in processing the innate visual threat, share similar downstream targets, their specific contribution to the process deserves further inquiry.

Consistent with the notion that the innate and learned aversive stimuli are conveyed through the LT inputs to the BLA, the activities of these two regions largely mirrored each other; however, noticeable differences were observed. For example, in line with previous studies on innate defensive response to predators odors (*Hayley et al., 2001*; *Hu et al., 2007*; *Do Monte et al., 2008*; *Liu et al., 2010*), in mice injected with propranolol, the defensive responses to the looming stimulus diminished significantly (*Figure 6B and E*); the stimulus-induced activity, however, was reduced only in the BLA (*Figure 6C*), with the response of the axons of the BLA-projecting LT neurons remaining unchanged (*Figure 6F*). This indicates that the looming stimulus conveyed through the thalamic input is essential but not sufficient to activate BLA neurons and trigger defensive responses, and the activation of β-adrenergic receptors by the release of norepinephrine is required. The neuromodulator may enhance the excitability of the pyramidal neurons directly by downregulating potassium channels in these neurons (*Faber et al., 2008*) or indirectly by reducing excitability of inhibitory neurons (*Tully et al., 2007*). Given the time-locked neuronal activity to the stimuli, the modulation of excitability, a comparably slow process, may be achieved through a tonic release of norepinephrine. As for the LT, to the best of our knowledge, there is little published work regarding its modulation by norepinephrine or even if β-adrenergic receptors are expressed in this region. Therefore, at this stage, we have no grounds to speculate about the ineffectiveness of propranolol in blocking the LT projection response to the looming stimulus. Here, we should point out that administration of propranolol prior to the conditioning or recall had no effect on freezing response. The literature on the role of norepinephrine in cued conditioning is mixed. While intraperitoneal injection of propranolol prior to a cued memory recall reduces freezing to the tone in rats (*Rodriguez-Romaguera et al., 2009*), the drug may reduce (*Leal Santos et al., 2021*) or have no effect (*Cain et al., 2004*) on the freezing response in mice. The differences in species or strains used or experimental parameters may contribute to the variability in the effect of the drug in freezing response.

Regarding the processing of the learned threat, again, we observed some differences between the LT projections to the BLA and the BLA itself. As animals learned the CS-US association, we observed an enhanced short-onset auditory response in the BLA (*Figure 5J*; *Quirk et al., 1995*). We did not detect a similar conditioning-correlated change in the activity of the LT axons (*Figure 3L*). It must be noted that because of our use of calcium indicators, we cannot exclude millisecond changes in the auditory response latency in the LT axons. However, previous works using sub-millisecond single-unit recording also showed similar patterns (*Barsy et al., 2020*; *Bordi and LeDoux, 1992*). The BLA also differed from the incoming LT projections in its response to the CS after an extensive extinction protocol. While activity of the LT axons returned to its preconditioning value, the CS-evoked activity of the BLA neurons after the extinction was significantly reduced compared to its value prior to the conditioning. This strongly suggests that other inputs to the BLA contribute to such a pronounced reduction. Feedforward inhibition of excitatory neurons in the BLA through synaptic potentiation (*Polepalli et al., 2010*) or dopaminergic modulation (*Bissière et al., 2003*) of the local inhibitory neurons may to some extent dampen the CS-evoked response after the extinction. Also, it has been proposed that norepinephrine can promote extinction through the activation of the infralimbic region, which in turn blunts the BLA activity (*Giustino and Maren, 2018*; *Uematsu et al., 2017*). It is pertinent

to mention that we have used an extensive multiple-session extinction training. Recently, it has been shown that such an extensive extinction protocol may involve a different mechanism by which the original fear memory is erased, and the extinguished CS becomes habituated (*An et al., 2017*).

Although not the focus of this work, we observed several intriguing physiological features during the conditioning and recall sessions. The conditioning increases the tone-induced activity in the BLA and reduces the response time onset (*Barsy et al., 2020*; *Bordi et al., 1993*). The increased response and decreased time onset were significantly more pronounced on the recall day. We observed a similar pattern in the LT input where there was a significant enhanced activity during the recall session, which was not visible during the conditioning. This is in line with previous studies using single-cell imaging in the BLA (*Grewe et al., 2017*). This significant change between the last trial of the conditioning and the first trial of the recall is puzzling. We speculate that a lack of a *detectable* increase in the CS-evoked activity at the later stages of the conditioning could be caused by a masking effect from a transient increase in firing rate during the conditioning. Although we did not observe a difference in the overall baseline activity during the conditioning (data not shown), downregulation as well as upregulation of the basal neuronal activity of subpopulations of neurons during the conditioning has been reported. This counteracting phenomenon could produce a net effect of no change in the baseline activity at the population level, while a subset of neurons with increased basal firing rate, possibly caused by enhanced excitability, may undergo plasticity. The plasticity within this population, however, will be masked during the conditioning by the transient increase in the basal firing rate. Alternatively, synaptic potentiation may occur at the dendritic compartments, which through local inhibitory circuits is uncoupled from somatic activity.

The LT-BLA pathway typically has been evaluated in relation to associative learnings (*Janak and Tye, 2015*; *Tye et al., 2008*), especially associative learned threats (*Barsy et al., 2020*; *Taylor et al., 2021*). Recent works on associative learned threats have particularly solidified the importance of associative plasticity in the LT (*Barsy et al., 2020*; *Taylor et al., 2021*). A recent study has further documented the critical role of the LT in processing different forms of innate threat (*Kang et al., 2022*). Our main aim in this work was to investigate the similarities and differences in processing an innate threat, which relies on pre-wired circuits, and a learned threat, which requires synaptic plasticity. By conducting a side-by-side comparison within the same animals, we not only gained new insights about shared and distinct features of processing the two forms of threats in the LT-BLA pathway, but also learned that, despite being monosynaptically connected, the LT and the BLA differ in important ways, as we detailed in our discussion. This provides new avenues for further investigation.

## Methods

**Key resources table**

| Reagent type (species) or resource | Designation | Source or reference | Identifiers | Additional information |
|---|---|---|---|---|
| Antibody | Anti-NeuN antibody (mouse) | Merck Millipore | MAB377 | 1:500 |
| Antibody | Anti-GFP antibody (rabbit) | Invitrogen | CAB4211 | 1:1000 |
| Antibody | Cy3 goat antimouse | Thermo Fisher Scientific | A10521 | 1:500 |
| Antibody | Alexa Fluor 488 (goat anti-rabbit) | Thermo Fisher Scientific | A-11008 | 1:1000 |
| Recombinant DNA reagent | AAV-5/2-hEF1αdlox- (pro)taCasp3_2A _TEVp(rev)-dlox | VVF | V185-5 | |
| Recombinant DNA reagent | AAV-1/2-hCMVchI-Cre | VVF | V36-1 | |
| Recombinant DNA reagent | AAV-5/2- mCaMKIIαhM4D(Gi)_mChe rry | VVF | V102-5 | |
| Recombinant DNA reagent | AAV-5/2- mCaMKIIαmCherry | VVF | V199-5 | |
| Recombinant DNA reagent | AAV-5/2-hSyn1- dlox-EGFP(rev)- dlox | VVF | V115-5 | |
| Recombinant DNA reagent | AAV-retro/2- hCMV-chI-Cre | VVF | V36-retro | |
| Recombinant DNA reagent | AAV-5/2-hSyn1- chI-jGCaMP7s | VVF | V406-5 | |

*Continued on next page*

*Continued*

| Reagent type (species) or resource | Designation | Source or reference | Identifiers | Additional information |
|---|---|---|---|---|
| Recombinant DNA reagent | AAV-5/2- mCaMKIIαjGCaMP8m | VVF | V630-5 | |
| Recombinant DNA reagent | AAV-5/2-hSyn1- chIChrimsonR_tdTo mato | VVF | V334-5 | |
| Chemical compound, drug | Propranolol hydrochloride | Merck | P0884 | |
| Chemical compound, drug | Sotalol hydrochloride | Merck | S0278 | |
| Chemical compound, drug | Clozapine noxide dihydrochloride (CNO watersoluble) | HelloBio | HB6149 | |
| Chemical compound, drug | Fentanyl | Hameln | 007007 | |
| Chemical compound, drug | Midazolam | Hameln | 002124 | |
| Chemical compound, drug | Medetomidine | VM Pharma | 087896 | |
| Chemical compound, drug | IsoFlo vet 100% | Zoetis | 37071/4000 | |
| Software | GraphPad Prism | GraphPad Software | Version 9 | |
| Software | ImageJ | National Institutes of Health | 1.53t | |
| Software | Doric Studio | Doric Lenses | 5.4.1.23 | |
| Software | MATLAB | MathWorks, Inc | R2021b | |
| Software, algorithm | PhotometrySignal-Analysis | This paper | https://github.com/NabaviLab-Git/Photometry-Signal-Analysis; *Nabavi Lab, 2022* | |
| Other | DAPI | Sigma | D9542 | 1:1000 |

## Animals

All the procedures were performed on C57BL/6JRJ wildtype (Janvier, France). Mice were naïve and acclimated to the vivarium for at least a week before the beginning of the experiment. The mice were 6–8 weeks old at the beginning of the experimental procedures. Animals were group-housed (3–4 per cage) with enriched conditions in a 12 hr light/dark cycle (the light switches on at 6 AM) with constant level of humidity and temperature (22 ± 1). Food and water were provided ad libitum. Behavioral experiments were conducted between 11 AM and 10 PM at Aarhus University at the Biomedicine department, Ole Worms Allé 8, Aarhus 8000. All the experimental procedures were conducted according to the Danish Animal Experiment Inspectorate.

## Stereotaxic surgery and virus expression

Mice were anesthetized using isoflurane (IsoFlo vet 100%, Zoetis), and standard surgical procedures were used to expose the skull. For most of the experiments, stereoscope lights were not used during the surgical procedures because they reduced the behavioral responses (*Figure 1—figure supplement 4*). Before the surgery, the mice were injected subcutaneously with buprenorphine 0.3 mg/mL (Temgesic, 0.1 mg/kg).

### Electrolytic-induced lesion of the BLA

Mice were lesioned bilaterally in the BLA at the following coordinates: anteroposterior (AP): –1.4 mm; mediolateral (ML): ±3.6 mm; dorsoventral (DV): –3.85/–4.1/–4.35 mm from the skull. The electrolytic lesion was performed with a concentric bipolar electrode (50691, Stoelting, USA) by delivering a constant direct current at each location (0.6 mA for 15 s). In another group of mice, the electrode was placed at the same location without delivering any current (sham surgery).

### Lesion of the BLA

Mice were bilaterally injected with a mixture of AAV-5/2-hEF1α-dlox-(pro)taCasp3_2A_TEVp(rev)-dlox (titer: $4.7 \times 10^{-12}$ vg/mL) in AAV-1/2-hCMV-chI-Cre (titer: $1.0 \times 10^{-13}$ vg/mL, ratio 7:1.5). The volume

of injection was 0.5 μL per hemisphere at the following coordinates AP: –1.6 mm; ML: ±3.45 mm; DV: –3.5/–4.1 mm from the skull. Control mice were injected with AAV-5/2-hEF1α-dlox-(pro)taCasp3_2A_TEVp(rev)-dlox at the same coordinates.

## Chemogenetics inhibition of the BLA

Mice were bilaterally injected with AAV-5/2-mCaMKIIα-hM4D(Gi)_mCherry (titer: 8.9 × 10⁻¹² vg/mL) or with AAV-5/2-mCaMKIIα-mCherry (titer: 6 × 10⁻¹² vg/mL, diluted 1:1 in PBS). The injection volume was 1 μL per hemisphere at the following coordinates for the BLA: AP: –1.6 mm; ML: ±3.45 mm; DV: –3.5/–4.1 mm from the skull.

## Lesion of the TeA and AuC

Mice were bilaterally injected with a mixture of AAV-5/2-hEF1α-dlox-(pro)taCasp3_2A_TEVp(rev)-dlox (titer: 4.7 × 10⁻¹² vg/mL) in AAV-1/2-hCMV-chI-Cre (titer: 1.0 × 10⁻¹³ vg/mL, ratio 7:1.5). The volume of injection was 0.2 μL per hemisphere at the following coordinates at AP: –2.85 mm; ML: ±4.44 mm; DV: –1.6 mm from the skull. Control mice were injected with AAV-5/2-hEF1α-dlox-(pro)taCasp3_2A_TEVp(rev)-dlox.

## Selective lesion of the LT-projecting neurons to the BLA

Animals were injected bilaterally in the LT with a mixture of AAV-5/2-hEF1α-dlox-(pro)taCasp3_2A_TEVp(rev)-dlox in AAV-5/2-hSyn1-dlox-EGFP(rev)-dlox titer: 1.1 × 10–13 vg/mL, ratio 7:2 at the following coordinates at AP: –3.15 mm; ML: ±1.85 mm; DV: –3.4/–3.5 mm from the skull. The same mice were injected bilaterally with AAV-retro/2-hCMV-chI-Cre (titer: 4.4 × 10⁻¹² vg/mL) in the BLA using the same coordinates mentioned earlier. One control group was injected with AAV-5/2-hEF1α-dlox-(pro)taCasp3_2A_TEVp(rev)-dlox in AAV-5/2-hSyn1-dlox-EGFP(rev)-dlox (ratio 7:2) in the LT. Another control group was injected with AAV-retro/2-hCMV-chI-Crein the BLA and with AAV-5/2-hSyn1-dlox-EGFP(rev)-dlox in the LT using the same dilution mentioned above.

## LT-BLA disconnection experiment

Mice were injected contralaterally in one LT and one LA (randomized hemispheres). In both locations, a mixture of AAV-5/2-hEF1α-dlox-(pro)ta-Casp3_2A_TEVp(rev)-dlox in AAV-1/2-hCMV-chI-Cre (ratio 7:1.5). The injection volume was 1 μL per hemisphere at the coordinates described previously. For the reversible disconnection experiment, mice were injected contralaterally in one LT and one BLA with AAV-5/2-mCaMKIIα-hM4D(Gi)_mCherry or with AAV-5/2-mCaMKIIα-mCherry (diluted 1:1 in PBS) in the BLA. Additional control groups were injected unilaterally with AAV-5/2-mCaMKIIα-hM4D(Gi)_mCherry in the LT or in the BLA.

## Fiber photometry experiments

Mice were injected unilaterally with AAV-5/2-hSyn1-chI-jGCaMP7s (titer: 7.7 × 10⁻¹² vg/ml) in the LT or with AAV-5/2-mCaMKIIα-jGCaMP8m (titer: 6.5 × 10⁻¹² vg/mL) in the BLA. The mice injected with jGCaMP8m in the BLA were injected in the LT with AAV-5/2-hEF1α-dlox-(pro)taCasp3_2A_TEVp(rev)-dlox in AAV-1/2-hCMV-chI-Cre (ratio 7:1.5). The injection volume was 0.5 μL per hemisphere at the previously mentioned coordinates. Control mice were injected with AAV-5/2-hEF1α-dlox-(pro)taCasp3_2A_TEVp(rev)-dlox at the same dilution. Mono fiber-optic cannula (200/300 um, NA 0.37) was implanted in the BLA at the following coordinates AP: –1.6 mm; ML: +3.48 mm; DV: –3.45 mm from the skull. The fiber-optic cannula was fixed to the skull with Superbond (SUN MEDICAL, Japan).

All the viral vectors were bought from the Viral Vector Facility (VVF) of the Neuroscience Center Zurich (ZNZ).

## Behavioral procedures

### Looming stimulus

The apparatus consisted of an open-top arena (37 × 40 × 19.5 cm) with a monitor (16 inches) placed on the top. No shelter was used (*Barbano et al., 2020*; *Shang et al., 2018*). The exposure to the looming stimulus was performed during the dark period (between 6 PM and 10 PM). Mice were placed in the center of the arena and explored freely for 8–10 min before exposure to the looming

stimulus. The looming stimulus consisted of an expanding black disk over a gray background, and it consisted of five repetitions, from 2° to 20° of visual angle (*Yilmaz and Meister, 2013*). The loom widens in 250 ms and remains at the same size for 250 ms with 2 s of pause between each loom. The experimenter was blind to the treatment and manually delivered the looming stimulus. The behavioral responses were recorded with a top camera (Phihong POE21U-1AF). All the animals were exposed at least 2–3 times to the looming stimulus, and only the one eliciting the greatest defensive response was analyzed. The defensive responses were analyzed automatically and manually using ANY-maze software (Stoelting, Ireland) for the 30 s following the onset of the looming stimulus. An experimenter, blind to the treatment, analyzed each mouse's freezing percentage, tail rattling, escape events, and rearing events. Freezing was defined as a complete lack of movements, except for the respiratory movements, that lasted for at least 1 s. Tail rattling was defined as an event in which the mouse moved the tail vigorously. Escape events were defined as a sharp increase in the locomotor speed three times greater than the average speed before the exposure to the looming stimulus. Rearing was defined as an event in which the mouse stood on its hindpaws, and they were quantified for the 30 s preceding (baseline) and following the looming stimulus presentation. The differential score was calculated by subtracting the rearing events following the looming stimulus subtracted by the rearing events during the baseline period. Note that positive differential score values indicate an increase in rearing events compared with the baseline period, whereas negative scores indicate a decrease in rearing events compared with the baseline period.

For the reversible disconnection experiment, we tested the effect of the inhibition of the LT-BLA pathway on the locomotion and on the anxiety levels. Locomotion was indicated by line crosses in the looming arena during the first 8 min of habituation. Anxiety-like behavior was measured as the percentage of time spent in the center part of the looming arena during the same time period.

For the fiber photometry experiments, a white looming stimulus was used as a neutral control stimulus (*Yilmaz and Meister, 2013*). It was presented in a pseudorandom order alternated with the black looming stimulus. The white looming stimulus presented the same repetitions and speed as the black looming stimulus.

## Aversive conditioning

The apparatus consisted of an open-top cage (24 × 20 × 30 cm) with metal floor bars placed in a soundproof cubicle (55 × 60 × 57 cm) (Ugo Basile, Italy). Two different behavioral protocols were used in this study. In one, the mice were conditioned by using five pairings consisting of 20 s, 5 kHz, sinewave tone (CS) co-terminating with a 2 s footshock (US) 0.6 mA. On the other hand, the animals were conditioned using four pairings consisting of 25 s, 7 kHz, sinewave tone co-terminating with 2 s foot shock 0.5 mA. After the conditioning, the animals stayed isolated for 10–15 min before returning to their home cage. Short-term (STM) and long-term memory (LTM) recall were assessed in a new context 2 hr or 24 hr after the conditioning, respectively. After 2 min of acclimation to the new context, the mice were presented to four or five CS presentations without the footshock. The intertrial interval ranged between 35 s and 120 s for both conditioning and testing sessions. The behavioral response was recorded by a top camera and/or side-view camera, and freezing was scored automatically by ANY-maze software. The freezing percentage indicates the time the mouse spent freezing during the CS presentation, divided by the CS length multiplied by 100. For the conditioning session, the freezing percentage is calculated for each CS presentation. For the recall session, the freezing percentage is the average of four CS presentations, with the only exception of the data shown in *Figure 1* and *Figure 1—figure supplements 1 and 2*, in which the freezing percentage is the average of five CS presentations. The pre-CS freezing percentage is calculated as the average of the freezing time during the 25 s before each CS presentation.

For the fiber photometry experiments, the mice underwent an extinction protocol for 2 d. On each day, the CS was played 12 times in the recall context without the footshock.

## Drugs

Propranolol hydrochloride (20 mg/kg, 500 µL; Merck, P0884) or sotalol hydrochloride (10 mg/kg, 500 µL; Merck, S0278) or clozapine N-oxide dihydrochloride (CNO water-soluble, HelloBio, HB6149; 10 mg/kg, 400 µL) or saline (0.9% NaCl, 400/500 µL) were administered by intraperitoneal injection.

After the injection, the mice were isolated for 30 min before the beginning of the experimental procedures.

## Electrophysiology recordings

As previously described, mice were injected with inhibitory DREADDs or mCherry in the BLA. The same cohort of mice was injected with undiluted AAV-5/2-hSyn1-chI-ChrimsonR_tdTomato (titer: 5.3 × 10E12 vg/mL) in the LT. After 4–5 wk of expression, the mice were anesthetized using 0.5 mg/kg FMM with the following mixture: 0.05 mg/ml of fentanyl ([Hameln, 007007] 0.05 mg/kg), plus 5 mg/mL of midazolam ([Hameln, 002124] 5 mg/kg), and 1 mg/mL of medetomidine (VM Pharma, 087896). After the induction of the anesthesia, the mice were placed on the stereotaxic frame. A 32-channel optoelectrode (Poly3, Neuronexus) was placed at the following coordinates: AP: 1.6 mm; ML: 3.48; DV: 3.5 ± 0.1. The neural data were amplified and digitized at 25 kHz. The input–output curve was recorded 30 min before and 1 hr after the CNO injection using a pulse of 0.5/1 ms, 638 nm.

At the end of the experiment, mice were sacrificed by cervical dislocation, and brains were extracted and kept in 10% formalin for 24 hr. Afterward, the brains were processed to confirm viral expression and electrode location.

## Fiber photometry recordings and analysis

All the recordings were performed with Doric fiber photometry system composed of an LED-driver, a fiber photometry console, and a Doric minicube with 460–490 nm for GCaMP excitation, 415 nm for isosbestic excitation, and 580–650 nm for optical stimulation [ilFMC5-G2_IE(400-410)_E(460-490)_F(500-540)_O(580-680)_S]. A low-autofluorescence patch cord (200 nm or 300 nm, 0.37 NA) and a pigtailed rotary joint (200 nm, 0.37 NA) were used. The latter were bleached for 5 hr before each experiment via a 473 nm laser at 15–17 mW. The GCaMP signal was amplified with a Doric amplifier with a 10× gain and recorded with Doric Neuroscience Studio software (version 5.4.1.23) at 11 kHz. The signal was downsampled at 120 Hz for analysis. Light power at the patch cord tip was set between 30 and 35 μW for 470 nm excitation.

For synchronization with the looming stimulus presentation, a National Instrument board (NI USB 6003) was used to time-stamp the looming stimulus presentation over the calcium signal. For synchronization of the tone and shock presentation, an input–output box connected to the aversive conditioning system was used.

After 4–5 wk of virus expression, mice were handled for 2–3 d before the beginning of the experiments. Before each recording, the fiber-optic cannula was cleaned with CleanClicker (Thorlab; USA). To reduce bleaching during the behavioral experiments, the habituation to the looming stimulus and the extinction to the tone were conducted without recording the GCaMP signal.

A customized MATLAB script was used for the analysis. All the traces with a sudden change in the isosbestic signal were discarded in the final analysis. The code used for the analysis is freely available at the following link: https://github.com/NabaviLab-Git/Photometry-Signal-Analysis (copy archived at *Nabavi Lab, 2022*). Briefly, the signals were downsampled to 120 Hz using local averaging. A first-order polynomial was fitted onto the data using the least-squares method. To calculate the relative change in fluorescence, the raw GCaMP signal was normalized using the fitted signal according to the following equation: deltaF/F = (GCaMP signal fitted signal)/(fitted signal). Behavioral events of interest were extracted and standardized using the mean and standard deviation of the baseline period. In all the plots, we used only individual trials for each animal.

The area under the curve (AUC) was calculated by using a built-in function in GraphPad Prism 9. All the peaks with a distance below 10% between the minimum and the maximum were discarded. For the AUC analysis for the looming stimulus, a baseline period of 2 s was considered and the first 2 s of the looming stimulus presentation. For the CS, we used the 1.5 s as the baseline period and first 2 s of the CS presentation. For the US, we analyzed the AUC for 0.5 s preceding and following the footshock presentation.

## Immunofluorescence

The mice were anesthetized with isoflurane and euthanized by cervical dislocation. The brains were harvested and stored for 24 hr in 10% formalin at room temperature. Then, the brains were sliced into 100–120-μm-thick slices in PBS on Leica Vibratome (VT1000S).

To visualize the extent of the lesion, the brains were stained for NeuN and GFP. Slices were permeabilized with PBS-Triton X 0.5% plus 10% of normal goat serum (NGS) and blocked in 10% bovine goat serum (BSA) for 90 min at room temperature. Subsequently, the slices were incubated with a mixture of anti-NeuN antibody mouse (Merck Millipore, MAB377; 1:500) and anti-GFP (Invitrogen, CAB4211, 1:1000) in PBS-Triton X 0.3%, 1% NGS, and 5% BSA and the incubation lasted for 72 hr at 4°C. At the end of the 72 hr incubation, the slices were washed three times in PBS. The slices were incubated in Cyanine 3 (Cy3) goat anti-mouse (Thermo Fisher Scientific, A10521, 1:500) and Alexa Fluor 488 goat anti-rabbit (Thermo Fisher Scientific, A-11008, 1:1000) in PBS-Triton X 0.3%, 1% NGS, and 5% BSA for 24 hr at 4°C. Nuclear staining was performed by using 1:1000 of DAPI (Sigma, D9542) for 30 min at room temperature. Brain slices were mounted on polysine glass slides with coverslips using Fluoromount G (Southern Biotech).

## Imaging and cell counting

Imaging was performed by using a virtual slide scanner (Olympus VS120, Japan). Tile images were taken by the whole brain slides by using ×10 (UPLSAPO 2 ×10/0,40) or ×20 objective (UPLSAPO ×20/0.75). The emission wavelength for Alexa 488 was 518 nm with 250 ms of exposure time. For Cy3, the emission wavelength was 565 nm with 250 ms of exposure time.

GFP-positive cells were counted manually by using ImageJ. The experimenter, blind to the treatment, defined a region of interest and performed the cell counting. The GFP-positive cells were quantified in the LT at AP: –3.40 mm from bregma, the region with highest density of GFP-positive cells in the control group. The area was defined by overlaying the atlas landmarks over the image. The area dimension was calculated by using the built-in function in ImageJ.

## Statistics

Statistical analyses were performed by using GraphPad Prism 9. All the data are represented as mean ± SEM, and they were tested for normality using Shapiro–Wilk and D'Agostino–Pearson normality test. If the data represented a normal distribution, a parametric test was used. The statistical methods and the corresponding p-values are reported in the figure legends. All the data were screened for outliers by using the ROUT test (Q = 0.5%).

## Acknowledgements

We thank R Malinow, J Piriz, F Ferenc Mátyás, J Lima, and members of the Nabavi laboratory for suggestions. We thank Jean-Charles Paterna and Viral Vector Facility (VVF) of the Neuroscience Center Zurich (ZNZ) for support. We thank Zachary Leamy for comments on the manuscript. This study was supported by Independent Research (DFF), Novo Nordisk Foundation (NNF16OC0023368), and AUFF NOVA grants to SN. Additionally, SN was supported by an ERC starting grant (22736), the Danish Research Institute of Translational Neuroscience (19958), and PROMEMO (Center of Excellence for Proteins in Memory funded by the Danish National Research Foundation) (DNRF133). Figures were created with BioRender.com.

## Additional information

### Funding

| Funder | Grant reference number | Author |
| --- | --- | --- |
| Danish Council for Independent Research | | Sadegh Nabavi |
| Novo Nordisk | NNF16OC0023368 | Sadegh Nabavi |
| AUFF NOVA | | Sadegh Nabavi |
| Danish Research Institute of Translational Neuroscience | 19958 | Sadegh Nabavi |

| Funder | Grant reference number | Author |
|---|---|---|
| Danish National Research Foundation | DNRF133 | Sadegh Nabavi |
| European Research Council | Starting grant 22736 | Sadegh Nabavi |

The funders had no role in study design, data collection and interpretation, or the decision to submit the work for publication.

## Author contributions

Valentina Khalil, Formal analysis, Validation, Investigation, Visualization, Methodology, Writing – original draft; Islam Faress, Conceptualization, Formal analysis, Validation, Investigation, Visualization, Methodology, Writing – original draft; Noëmie Mermet-Joret, Validation, Investigation, Visualization, Methodology, Writing – review and editing; Peter Kerwin, Software, Writing – review and editing; Keisuke Yonehara, Conceptualization, Writing – review and editing; Sadegh Nabavi, Conceptualization, Resources, Formal analysis, Supervision, Funding acquisition, Validation, Writing – original draft, Project administration, Writing – review and editing

## Author ORCIDs

Valentina Khalil ⓘ http://orcid.org/0000-0001-5928-8596
Islam Faress ⓘ http://orcid.org/0009-0000-2218-9180
Peter Kerwin ⓘ http://orcid.org/0000-0001-8792-8626
Sadegh Nabavi ⓘ https://orcid.org/0000-0002-3940-1210

## Ethics

All the animal expermentations performed here were reviewed and approved by Danish Animal Experiment Inspectorate (permit number 2020-15-0201-00421).

## Decision letter and Author response

Decision letter https://doi.org/10.7554/eLife.85459.sa1
Author response https://doi.org/10.7554/eLife.85459.sa2

## Additional files

### Supplementary files

• MDAR checklist

### Data availability

All data generated or analysed during this study are included in the manuscript and supporting file; source data for all the figures are deposited at Dyrad and available at https://doi.org/10.5061/dryad.dbrv15f54 The codes generated for this work are available on GitHub at https://github.com/NabaviLab-Git/Photometry-Signal-Analysis (copy archived at *Nabavi Lab, 2022*).

The following dataset was generated:

| Author(s) | Year | Dataset title | Dataset URL | Database and Identifier |
|---|---|---|---|---|
| Nabavi SS | 2023 | Distinct representations of innate and learned threats within the thalamic-amygdala pathway | https://doi.org/10.5061/dryad.dbrv15f54 | Dryad Digital Repository, 10.5061/dryad.dbrv15f54 |

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
