## [Editor Report]

This study presents valuable insights into the circuits that are common for innate and acquired threats. The evidence supporting the conclusions is convincing, and the use of state-of-the-art methodology for the study of neural circuits, including chemogenetics, optogenetics, and fiber photometry, is appropriate. This work will be of interest to neuroscientists studying defensive behaviors as well as those in the field of multisensory thalamic integration.

---

## [Decision Letter]

**Decision letter after peer review:**

Thank you for submitting your article "Distinct representations of innate and learned threats within the thalamic-amygdala pathway" for consideration by *eLife*. Your article has been reviewed by 3 peer reviewers, and the evaluation has been overseen by a Reviewing Editor and Kate Wassum as the Senior Editor. The reviewers have opted to remain anonymous.

Essential revisions:

We collectively agree that the authors' report of a role for projections from the lateral thalamus to the basolateral amygdala in innate defensive responses to visual stimuli could be an important contribution to neurosciences. However, following a consultation session between the Reviewing Editor and the Reviewers several concerns were raised regarding the interpretation of the data and the methods employed. Based on these, we have enumerated the following recommendations that we encourage the authors to carefully address. In addition, the assessment from each of the reviewers is included below. With your revision, please include a point-by-point response to each reviewer comment.

1) All three Reviewers and the Reviewing Editor agree that proper histological characterization of the lateral thalamic-amygdalar pathway studied by the authors is required. This is critical given that although the authors solely refer to MGN there appears to be labeling in regions like PIL/SPFp and SG which are known to project to BLA and whose contributions to auditory fear conditioning are well established. Please attend to the authors specific suggestions. It would also help to have schematics of viral expression and placement maps showing spread and placements for all subjects for each experiment.

2) The Reviewers raise several methodological concerns that the authors should carefully consider. Note that addressing some of these may require additional experiments (e.g., circuit manipulations during memory retrieval).

3) In general, the Reviewers feel that the authors should discuss the interpretation of their data in further detail.

4) Please ensure your manuscript complies with the *eLife* policies for statistical reporting: https://reviewer.elifesciences.org/author-guide/full "Report summary statistics (e.g., t, F values) and degrees of freedom, exact p-values, and 95% confidence intervals wherever possible. These should be reported for all key questions and not only when the p-value is less than 0.05.

5) Please include a key resource table.

*Reviewer #1 (Recommendations for the authors):*

– The paper would be strengthened by showing that optogenetic inhibition of the MGM-BLA pathway precisely during auditory tone or looming stimulus presentation decreases freezing. It is odd that the authors used disconnections and lesions instead of projection optogenetic inhibition, which would provide the authors with much-needed temporal resolution in their manipulations.

– It is unclear if the MGM pathway is necessary only for timed stimulus-evoked freezing or whether this circuit generally alters the defensive state. Measuring the loss of function effects on open-field anxiety behavior would be informative in this regard.

– Not all figures contain photos of representative histology. Please add histology to all figures that are missing it. Please show that MGM virus expression did not spread into the visual lateral geniculate nucleus

– The authors show that the defensive response induced by the looming stimulus rapidly habituates, as shown by other groups. The authors should discuss why this phenomenon occurs, as it is highly unusual. Mice do not rapidly habituate to any other innate fear or anxiety-inducing stimuli, such as predators or open spaces.

*Reviewer #2 (Recommendations for the authors):*

Additional specific comments to the authors:

a) Given that the regions (and a significant proportion of individual neurons) that project to the BLA also project to the neocortex, and in turn neocortex projects strongly to the amygdala as well, based on the presented data it is possible to interpret that the observed role of the MGN-BLA in threat processing is not merely direct as concluded in the study but could also be indirect through the cortex. The function of these two pathways to BLA is a major topic in the field, and the study would thus benefit from a discussion about this alternative interpretation.

b) A main claim of the authors is that mice with disrupted MGN-BLA fail to switch from exploration to freezing. May the authors quantify this directly to support this claim?: for example, rearing before vs. after stimulus onset. On similar lines, is it possible that instead of decreasing freezing, disrupting the pathway is increasing escape responses? It does not seem to be the case by looking at the supplementary figures, but making this explicit in the text would help.

c) The freezing behaviour and pathway activity mostly go hand-in-hand, but it would be relevant to discuss: 1) at the end of the conditioning phase freezing is increased but not the CS evoked response in MGN axons. 2) the MGN-BLA activity decays with habituation and extinction, but these are different cognitive processes. What is the interpretation in both cases?

d) The white looming stimulus is a good control. Have the authors considered using a nonaversive black stimulus?

e) Authors state that the time locking of the calcium signals to stimuli is evident, but it would be good to support it with a quantification. Why does the BLA signal have a shorter delay than the MGN axons signal? This is not expected.

f) Some methodological issues are not clarified and generate confusion:

1) When is freezing or rearing measured with respect to stimuli presentation (looming, sounds, foot-shocks)?

2) Are we looking at averages of trials in the data or individual trials?

3) How is habituation to looming stimuli defined? Is there habituation in the first session already?

4) Why is memory recall sometimes 2h and others 24 hs after conditioning throughout the work?

5) In the selective lesion experiments, where were GFP+ neurons counted in the anterior-posterior axis, and how were the areas defined?

6) How were AUC and z-score quantified?

g) On similar lines the clarity of the figures should be improved:

1) Define unit in rearings plots. Is it a number of rearings?

2) For fiber-photometry recordings and ethograms, please indicate the duration of stimuli in the plot (looming, sounds, foot-shocks), and label it on the plot. It would also help to use the same time scale across similar plots, e.g. in 3H-J and 5E-G.

3) Figure 1a: can you show the placement of the recording electrode?

4) Figure 1b: is it possible that the two first traces without and with CNO are identical? The term "baseline" is confusing here; maybe you could say "before CNO". State the meaning of fEPSP.

5) Figure 1c. What is presented in the picture exactly? the mCherry expression looks way outside the BLA. Is it maybe not the best representative image?

6) Figure 1E legend mistake: "rearing is reduced".

7) For figure 1h and all other equivalent quantifications in the manuscript, what are baseline freezing levels (before sound presentation) in memory recall? It is important to show that freezing is evoked by the sound.

8) Figure 2c. I stumbled upon this picture. Green neurons are visible in the MGNv and in the dentate gyrus of the hippocampus, regions that do not project to BLA. How is it possible?

9) Font sizes are sometimes too small and difficult to read.

h) The design of the propranolol experiment makes the interpretation of results difficult. An alternative explanation to the one offered by the authors is that propranolol impairs habituation and the concomitant decrease in MGN axons activity.

i) Typo in the main text: figure S10D-I is cited instead of S6D-I.

j) Methods indicate that "all the data were screened for outliers". What happened if outliers were found?

k) Given that the main findings are about similarities, rather than differences, between innate and learnt threat, maybe the title could be modified.

*Reviewer #3 (Recommendations for the authors):*

1. The viral infection in the BLA at least from Figure 1C seems to have some spread to the surrounding regions, especially the central amygdala.

Since the central amygdala is known to play a predominant role in the generation of freezing responses, it would be important to exclude that viral leakage to this region is the basis of the effects on freezing described in Figure 1D-H. A detailed representation of the spread of each infection is needed. Lesions and Caspase3 experiments also show a similar spread to the CEA

2. The authors should provide a much more detailed description of fiber photometry analysis. For example, they should specify in which time interval the AUC was calculated.

3. A more detailed analysis of fiber photometry signals may add important information. The AUC is certainly a good starting point and tells us there is a statistically significant activation. From the plots it looks like the responses ramp up quite slowly. I wonder if this is not related to the defensive responses induced by looming rather than by the looming itself. To disentangle this, it would be important to align traces to freezing bouts inside and outside looming stimuli.

4. A more detailed analysis of the timing of photometry responses in Figure 3 and Figure 5 would add a lot to the manuscript.

5. In Figure 5 A-G the authors nicely show that lesions of the MGN abolish looming stimulus-mediated responses observed in the BLA. This result in itself is clear and very convincing. However, I have some doubts about the control group. The control group shows activity in the BLA in response to looming stimuli. From both the average traces in 5C and the heat maps in 5E, it looks like the delay between the start of the looming stimulus and the activity increase is about 250 ms, while in Figure 3F-H the delay is clearly longer (500 ms). If the MGN is upstream of the BLA, how is it possible that it has a longer delay?

6. Similarly, in Figure 5J the delays of the responses for the control group are puzzling in many aspects. First, in the recall group, again the delay of BLA neurons is much shorter than the one described for MGN neurons, which should be their inputs according to the author's hypothesis. Second, the delay of responses changes a lot between recall and the first and last CS groups. This deserves a detailed analysis and elaboration on what are the possible mechanisms at the basis of this in the discussion. Third, while in the MGN CS responses during conditioning and extinction are comparable, in the BLA neurons display CS responses during extinction that are much lower than the ones during conditioning. This is a marked discrepancy between MNG and BLA activity that suggests that other BLA inputs contribute to the active inhibition of BLA neurons during extinction. This may indeed be somehow related to adrenergic inputs which also show functional differences in Figure 6. This should be elaborated on in the discussion.

7. It is nice to see that the results are very much in line with what was recently reported by Kang and colleagues in 2022 who showed that SPFp (a part of the MGN) responds to unconditioned cues including looming stimuli. Similarly, Taylor et al. 2021 showed that MGB neurons projecting to the BLA respond to tones and are modulated by fear conditioning. Nevertheless, these two studies impinge on the novelty of the findings here. Can the authors help the reviewers clarify what are the important novel findings of this study?

---

## [Author Response]

Essential revisions:We collectively agree that the authors' report of a role for projections from the lateral thalamus to the basolateral amygdala in innate defensive responses to visual stimuli could be an important contribution to neurosciences. However, following a consultation session between the Reviewing Editor and the Reviewers several concerns were raised regarding the interpretation of the data and the methods employed. Based on these, we have enumerated the following recommendations that we encourage the authors to carefully address. In addition, the assessment from each of the reviewers is included below. With your revision, please include a point-by-point response to each reviewer comment.1) All three Reviewers and the Reviewing Editor agree that proper histological characterization of the lateral thalamic-amygdalar pathway studied by the authors is required. This is critical given that although the authors solely refer to MGN there appears to be labeling in regions like PIL/SPFp and SG which are known to project to BLA and whose contributions to auditory fear conditioning are well established. Please attend to the authors specific suggestions. It would also help to have schematics of viral expression and placement maps showing spread and placements for all subjects for each experiment.2) The Reviewers raise several methodological concerns that the authors should carefully consider. Note that addressing some of these may require additional experiments (e.g., circuit manipulations during memory retrieval).3) In general, the Reviewers feel that the authors should discuss the interpretation of their data in further detail.4) Please ensure your manuscript complies with the eLife policies for statistical reporting: https://reviewer.elifesciences.org/author-guide/full "Report summary statistics (e.g., t, F values) and degrees of freedom, exact p-values, and 95% confidence intervals wherever possible. These should be reported for all key questions and not only when the p-value is less than 0.05.5) Please include a key resource table.Reviewer #1 (Recommendations for the authors):– The paper would be strengthened by showing that optogenetic inhibition of the MGM-BLA pathway precisely during auditory tone or looming stimulus presentation decreases freezing. It is odd that the authors used disconnections and lesions instead of projection optogenetic inhibition, which would provide the authors with much-needed temporal resolution in their manipulations.

In our previous email to the editors, we mentioned work by Kang et al., 2022 (See Figure 5 and Figure 6 in Kang et al), which address this in a slightly different form. In the new revision, we refer to this experiment.

– It is unclear if the MGM pathway is necessary only for timed stimulus-evoked freezing or whether this circuit generally alters the defensive state. Measuring the loss of function effects on open-field anxiety behavior would be informative in this regard.

We measured the effects of inhibition of the LT-BLA pathway in mice injected contralaterally with hM4Di and CNO on locomotion and anxiety-related behaviors. These data are now included in Figure 4—figure supplement 2A.

– Not all figures contain photos of representative histology. Please add histology to all figures that are missing it. Please show that MGM virus expression did not spread into the visual lateral geniculate nucleus

In the corresponding supplementary figures, we show the maps of viral expression and optic fiber location for all the mice. In addition, we included a representative image to show that injection in the LT (MGN) does not spread into the LGN (Figure 4—figure supplement 1). As seen, there is no detectable cell body in the LGN. We observe axonal labeling in the auditory cortex and in the Zona Incerta, regions receiving projections from the LT.

– The authors show that the defensive response induced by the looming stimulus rapidly habituates, as shown by other groups. The authors should discuss why this phenomenon occurs, as it is highly unusual. Mice do not rapidly habituate to any other innate fear or anxiety-inducing stimuli, such as predators or open spaces.

In the new revision, we discuss this phenomenon: (A recent study has shown that upon repeated exposure … Further studies are necessary to test this possibility.)

Reviewer #2 (Recommendations for the authors):Additional specific comments to the authors:a) Given that the regions (and a significant proportion of individual neurons) that project to the BLA also project to the neocortex, and in turn neocortex projects strongly to the amygdala as well, based on the presented data it is possible to interpret that the observed role of the MGN-BLA in threat processing is not merely direct as concluded in the study but could also be indirect through the cortex. The function of these two pathways to BLA is a major topic in the field, and the study would thus benefit from a discussion about this alternative interpretation.

We performed a new experiment showing that inactivation of the cortical region does not interfere with animals’ ability to produce defensive responses to looming stimulus (Figure 2—figure supplement 2).

b) A main claim of the authors is that mice with disrupted MGN-BLA fail to switch from exploration to freezing. May the authors quantify this directly to support this claim?: for example, rearing before vs. after stimulus onset. On similar lines, is it possible that instead of decreasing freezing, disrupting the pathway is increasing escape responses? It does not seem to be the case by looking at the supplementary figures, but making this explicit in the text would help.

We measured the differential score for rearing events before and after the looming stimulus presentation. These data are now included in Figure 1—figure supplement 2G, Figure 2—figure supplement 1B, and Figure 4—figure supplement 2H. Information about how the differential score was calculated is now included in the method section.

Mice with compromised LT-BLA pathway do not show freezing or escape behavior: (More specifically, upon exposure to an innate visual threat, animals with a compromised LT-BLA pathway fail to switch from exploratory to defensive behaviors, displaying neither freezing nor escape reaction)

c) The freezing behaviour and pathway activity mostly go hand-in-hand, but it would be relevant to discuss: 1) at the end of the conditioning phase freezing is increased but not the CS evoked response in MGN axons. 2) the MGN-BLA activity decays with habituation and extinction, but these are different cognitive processes. What is the interpretation in both cases?

In the revised manuscripts we have discussed both points: (This significant change between the last trial …which through local inhibitory circuits is uncoupled from somatic activity.); (Although reduced defensive response to an innately aversive…Further investigations in this line may have fundamental and translational value.)

d) The white looming stimulus is a good control. Have the authors considered using a nonaversive black stimulus?

We have not tested a non-aversive black stimulus. We agree with the reviewer that the use of a non-aversive black stimulus, such as an expanding black stimulus illuminated from bottom, is a valuable control. We will consider this in our future work.

e) Authors state that the time locking of the calcium signals to stimuli is evident, but it would be good to support it with a quantification. Why does the BLA signal have a shorter delay than the MGN axons signal? This is not expected.

We quantified the time-to-peak of GCaMP signals in response to looming and CS stimuli. The quantification is included in the legends for figures 3 and 5.

The difference in the delay can be attributed to the type of calcium indicators we used in the two different regions. GCaMP8s, which was used in the LT (MGN) projections, is suited to detect smaller signals produced by axonal activity, has a higher S/N but a slower rise time whereas GCaMP8m used in the BLA neurons has a faster rise time (Zhang et al., 2023).

f) Some methodological issues are not clarified and generate confusion:1) When is freezing or rearing measured with respect to stimuli presentation (looming, sounds, foot-shocks)?

We changed the relevant Y-axis to indicate in which period the freezing was measured. We measured rearing only during the looming sessions (please see methods). In the method section, under the heading Behavioral procedures we define the period used to measure freezing and rearing.

2) Are we looking at averages of trials in the data or individual trials?

In the updated methods section, we clarify whether the data represent individual or averaged trials.

3) How is habituation to looming stimuli defined? Is there habituation in the first session already?

Habituation was defined as a reduction in all measured defensive responses after repeated exposure to the stimulus. We observed habituation occurs within the first session (data not shown), but the phenomenon was more pronounced in the second session (Figure 1—figure supplement 1B).

4) Why is memory recall sometimes 2h and others 24 hs after conditioning throughout the work?

We used a 2-hr versus a 24-hr recall to assess the effect of our manipulations on short-term and long-term memory.

5) In the selective lesion experiments, where were GFP+ neurons counted in the anterior-posterior axis, and how were the areas defined?

This is now clarified in the method section.

6) How were AUC and z-score quantified?

This is now clarified in the method section.

g) On similar lines the clarity of the figures should be improved:1) Define unit in rearings plots. Is it a number of rearings?

The reported counts are the number of rearing events. We changed the y-axis in the relevant figures to clarify this.

2) For fiber-photometry recordings and ethograms, please indicate the duration of stimuli in the plot (looming, sounds, foot-shocks), and label it on the plot. It would also help to use the same time scale across similar plots, e.g. in 3H-J and 5E-G.

We included the duration of the CS and US epochs in the main figures. Also, the onset and offset time of the looming stimulus is shown in the ethograms. We used different scales for GcaMP7s and GcaMP8m reflecting their different kinetics.

3) Figure 1a: can you show the placement of the recording electrode?

A representative image of the recording site is now included in Figure 1—figure supplement 2A.

4) Figure 1b: is it possible that the two first traces without and with CNO are identical? The term "baseline" is confusing here; maybe you could say "before CNO". State the meaning of fEPSP.

Although they appear identical, the first two traces in figure 1b represent before and after CNO application. The light-evoked response during the whole recording period was stable. For clarity, we now overlay the standard error of the mean as a shaded area on the traces (figures 1B and 4D).

fEPSP is now defined.

5) Figure 1c. What is presented in the picture exactly? the mCherry expression looks way outside the BLA. Is it maybe not the best representative image?

In the revised manuscript, we replaced this with another image. In the majority of the cases, we observed either a lateral (cortical) or a medial (striatal) spread of the virus (Figure 1—figure supplement 2B). An experimenter who was blind to the treatments (hM4di-CNO, mCherry-CNO, and hM4di-Vehicle) and to the behavioral results evaluated the histological samples. Accordingly, mice lacking expression in the BLA or showing major leakage outside the BLA were excluded regardless of the behavioral outcome. We must note that it is technically challenging to obtain AAV expression that is confined within a deep brain region such as the BLA. Our results from the disconnection experiment, in which only one BLA and the contralateral LT were silenced with hM4di confirm that the LT-BLA pathway is necessary for innate and learned threat processing. In this experiment, the spared (unsilenced) BLA and contralateral LT serve as a control for off-target silencing of neighboring regions that occurred in silencing the BLA experiment.

6) Figure 1E legend mistake: "rearing is reduced".

Thank you, this is now corrected.

7) For figure 1h and all other equivalent quantifications in the manuscript, what are baseline freezing levels (before sound presentation) in memory recall? It is important to show that freezing is evoked by the sound.

Quantification of the pre-CS freezing is now shown in Figure 1—figure supplement 2I, Figure 2—figure supplement 1E and Figure 4—figure supplement 2J.

8) Figure 2c. I stumbled upon this picture. Green neurons are visible in the MGNv and in the dentate gyrus of the hippocampus, regions that do not project to BLA. How is it possible?

It is probably due to some leakage while retracting the pipette after the virus injections. However, the labeling observed is highly sparse (1 to 2 cells). This is unlikely to confound the interpretation of the results.

9) Font sizes are sometimes too small and difficult to read.

We increased the font size in the relevant parts.

h) The design of the propranolol experiment makes the interpretation of results difficult. An alternative explanation to the one offered by the authors is that propranolol impairs habituation and the concomitant decrease in MGN axons activity.

In the revised manuscript, we discuss these scenarios: (Additionally, while mice injected with propranolol…These studies may provide mechanistic insight into the habituation that we have observed here.); (This indicates that the looming stimulus conveyed through the thalamic input…blocking the LT projection response to the looming stimulus.).

i) Typo in the main text: figure S10D-I is cited instead of S6D-I.

Thank you, this is now corrected.

j) Methods indicate that "all the data were screened for outliers". What happened if outliers were found?

We did not find any outliers in our data set.

k) Given that the main findings are about similarities, rather than differences, between innate and learnt threat, maybe the title could be modified.

The title has been changed to “Subcortico-amygdala pathway processes innate and learned threats”.

Reviewer #3 (Recommendations for the authors):1. The viral infection in the BLA at least from Figure 1C seems to have some spread to the surrounding regions, especially the central amygdala.Since the central amygdala is known to play a predominant role in the generation of freezing responses, it would be important to exclude that viral leakage to this region is the basis of the effects on freezing described in Figure 1D-H. A detailed representation of the spread of each infection is needed. Lesions and Caspase3 experiments also show a similar spread to the CEA2. The authors should provide a much more detailed description of fiber photometry analysis. For example, they should specify in which time interval the AUC was calculated.

As requested, we have provided a detailed description of fiber photometry analysis in the method section.

3. A more detailed analysis of fiber photometry signals may add important information. The AUC is certainly a good starting point and tells us there is a statistically significant activation. From the plots it looks like the responses ramp up quite slowly. I wonder if this is not related to the defensive responses induced by looming rather than by the looming itself. To disentangle this, it would be important to align traces to freezing bouts inside and outside looming stimuli.

We have aligned the traces to freezing periods during the looming stimulus and the CS presentation in the recall session (Author response image 1). In many instances, freezing outlasted the looming stimulus. Mice did not initiate freezing behavior outside the stimulus periods.

Author response mage 1 shows freezing events aligned to the looming presentation (left) and to the first CS in the recall session (right) in mice expressing GCaMP8m in the BLA (Figure IA) or GCaMP7s in the LT (MGN) axons (figure IB). As shown, there is no detectable correlation between BLA and the LT activity and freezing onsets.

**Author response image 1. sa2fig1:** 

4. A more detailed analysis of the timing of photometry responses in Figure 3 and Figure 5 would add a lot to the manuscript.

In the legends for figures 3 and 5, we now add the values for time-to-peak of GCaMP signals in response to different stimuli.

5. In Figure 5 A-G the authors nicely show that lesions of the MGN abolish looming stimulus-mediated responses observed in the BLA. This result in itself is clear and very convincing. However, I have some doubts about the control group. The control group shows activity in the BLA in response to looming stimuli. From both the average traces in 5C and the heat maps in 5E, it looks like the delay between the start of the looming stimulus and the activity increase is about 250 ms, while in Figure 3F-H the delay is clearly longer (500 ms). If the MGN is upstream of the BLA, how is it possible that it has a longer delay?

The difference in the delay can be attributed to the type of calcium indicators we used in the two different regions. GCaMP8s, which was used in the LT (MGN) projections, is suited to detect smaller signals produced by axonal activity. This indicator has a higher S/N but a slower rise time. GCaMP8m used in the BLA neurons, on the other hand, has a faster rise time (Zhang et al., 2023).

6. Similarly, in Figure 5J the delays of the responses for the control group are puzzling in many aspects. First, in the recall group, again the delay of BLA neurons is much shorter than the one described for MGN neurons, which should be their inputs according to the author's hypothesis.

Please see our response above.

Second, the delay of responses changes a lot between recall and the first and last CS groups. This deserves a detailed analysis and elaboration on what are the possible mechanisms at the basis of this in the discussion.

We have re-analyzed our data, for example, by using the pre-CS1 period as the baseline for all the CSs, but results remain the same (data not shown). In the section Discussion, we offer some possible explanations for this phenomenon: (This significant change between the last trial of the conditioning…which through local inhibitory circuits is uncoupled from somatic activity.)

Third, while in the MGN CS responses during conditioning and extinction are comparable, in the BLA neurons display CS responses during extinction that are much lower than the ones during conditioning. This is a marked discrepancy between MNG and BLA activity that suggests that other BLA inputs contribute to the active inhibition of BLA neurons during extinction. This may indeed be somehow related to adrenergic inputs which also show functional differences in Figure 6. This should be elaborated on in the discussion.

We have discussed this in the revised manuscript: (The BLA also differed from the incoming LT projections in its response… by which the original fear memory is erased and the extinguished CS becomes habituated (An et al., 2017)).

7. It is nice to see that the results are very much in line with what was recently reported by Kang and colleagues in 2022 who showed that SPFp (a part of the MGN) responds to unconditioned cues including looming stimuli. Similarly, Taylor et al. 2021 showed that MGB neurons projecting to the BLA respond to tones and are modulated by fear conditioning. Nevertheless, these two studies impinge on the novelty of the findings here. Can the authors help the reviewers clarify what are the important novel findings of this study?

In the revised manuscript, we put in perspective our findings to those from the labs of Dr. Mátyás, Dr. Han, and Dr. Gründemann: (Recents works on associative learned threats…This provides new avenues for further investigation.)